# Differential effects of light and feeding on circadian organization of peripheral clocks in a forebrain *Bmal1* mutant

**Mariko Izumo[1†], Martina Pejchal[2†], Andrew C Schook[2,3], Ryan P Lange[2], Jacqueline A Walisser[4], Takashi R Sato[5,6], Xiaozhong Wang[7], Christopher A Bradfield[4], Joseph S Takahashi[1,8]***

[1]Department of Neuroscience, University of Texas Southwestern Medical Center, Dallas, United States; [2]Department of Neurobiology, Northwestern University, Evanston, United States; [3]Howard Hughes Medical Institute, Northwestern University, Evanston, United States; [4]McArdle Laboratory for Cancer Research, University of Wisconsin, Madison, United States; [5]Werner Reichardt Centre for Integrative Neuroscience, University of Tübingen, Tübingen, Germany; [6]JST, PRESTO, University of Tübingen, Tübingen, Germany; [7]Department of Molecular Biosciences, Northwestern University, Evanston, United States; [8]Howard Hughes Medical Institute, University of Texas Southwestern Medical Center, Dallas, United States

**Abstract** In order to assess the contribution of a central clock in the hypothalamic suprachiasmatic nucleus (SCN) to circadian behavior and the organization of peripheral clocks, we generated forebrain/SCN-specific *Bmal1* knockout mice by using floxed *Bmal1* and pan-neuronal Cre lines. The forebrain knockout mice showed >90% deletion of BMAL1 in the SCN and exhibited an immediate and complete loss of circadian behavior in constant conditions. Circadian rhythms in peripheral tissues persisted but became desynchronized and damped in constant darkness. The loss of synchrony was rescued by light/dark cycles and partially by restricted feeding (only in the liver and kidney but not in the other tissues) in a distinct manner. These results suggest that the forebrain/SCN is essential for internal temporal order of robust circadian programs in peripheral clocks, and that individual peripheral clocks are affected differently by light and feeding in the absence of a functional oscillator in the forebrain.

**\*For correspondence:** joseph. takahashi@utsouthwestern.edu

†These authors contributed equally to this work

**Competing interests:** The authors declare that no competing interests exist.

## Introduction

The suprachiasmatic nucleus (SCN) in the hypothalamus is the primary regulator of daily rhythms of behavior and physiology in mammals (*Meijer and Rietveld, 1989*; *Ralph et al., 1990*; *Sujino et al., 2003*; *Welsh et al., 2010*), yet the majority of tissues in the body possess autonomous cellular oscillators (*Yamazaki et al., 2000*; *Abe et al., 2002*; *Nagoshi et al., 2004*; *Welsh et al., 2004*; *Yoo et al., 2004*; *Abraham et al., 2005*). The circadian clock in the SCN as well as in other mammalian cells is composed of interacting positive- and negative-transcriptional and post-translational feedback loops involving *Clock* and *Bmal1* transcription factors and their target genes, *Period* (*Per1, 2,* and *3*) and *Cryptochrome* (*Cry1* and *2*) (*King et al., 1997*; *Gekakis et al., 1998*; *Hogenesch et al., 1998*; *Kume et al., 1999*; *Lee et al., 2001*; *Lowrey and Takahashi, 2004, 2011*; *Huang et al., 2012*; *Mohawk et al., 2012*). Extensive evidence suggests that the SCN controls not only behavioral rhythms but also the circadian programs of peripheral tissues (*Yoo et al., 2004*; *Guo et al., 2005, 2006*; *Saini et al., 2013*; *Yamaguchi et al., 2013*). However, much remains unclear regarding the relationship

**eLife digest** Jet lag is a common experience when flying long distance. This disorientating phenomenon occurs when our internal 'body clock' remains set to the time zone where the plane departed and fails to reset to the new local time.

Our internal clock actually consists of a series of clocks—each of which is based upon groups of genes that are switched on and off at different times of the day and night. There is a master clock in our brain and a series of peripheral clocks in our other organs and tissues. The master clock is thought to coordinate the peripheral clocks, which in turn control the fluctuating activity of a specific organ in response to the time of day.

To further investigate the master clock, a typical approach has been made to disable it by deleting the genes for its components. But some of these deletions can cause abnormalities in mice and some are lethal. To get around these problems, Izumo, Pejchal et al. have devised a way to delete a molecular component of the master clock only in the mouse's brain.

Izumo, Pejchal et al. used this approach to specifically disable the mouse's master clock and, unlike mice that completely lack the *Bmal1* gene, mice with the brain-specific deletion were as healthy and lived as long as normal mice. A molecular probe was used to monitor the peripheral clocks in different organs and tissues of these mutant mice, and revealed that, without a working master clock, the peripheral clocks were no longer synchronized. Izumo, Pejchal et al. found that the lost synchrony could be partially restored by training the mice to adapt to cycles of light and dark and feeding schedules.

Following on from the work of Izumo, Pejchal et al., one of the next challenges is to understand how the master clock communicates with the peripheral clocks in different organs and tissues around the body.

between the central clock and peripheral oscillators, which can be entrained by many different stimuli (*Stokkan et al., 2001*; *Buhr et al., 2010*; *Saini et al., 2012*). To clarify the contribution of a central clock to the circadian rhythms of peripheral tissues, we sought to find a way to disable the molecular oscillators in the brain without affecting the genetic components in the rest of the body.

Traditionally, a functional role of a gene has been investigated by a gene targeting strategy, which eliminates the gene from embryonic stem cells so that it is inactivated ubiquitously in mice. While this approach provides a powerful method to study the function of genes in vivo, it cannot be applied to genes that affect developmental or metabolic processes crucial for survival. Moreover, even if the knockout mice survive to adulthood, some lines suffer from systemic conditions and hence require special handling in experiments. For instance, although a global knock out of *Bmal1* ($Bmal1^{-/-}$) completely eliminates circadian rhythms (*Bunger et al., 2000*), the animals suffer from morbid conditions including arthropathy (*Bunger et al., 2005*), sterility (*Alvarez et al., 2008*; *Boden et al., 2010*), defects in glucose homeostasis (*Rudic et al., 2004*; *Lamia et al., 2008*; *Marcheva et al., 2010*), premature aging, and decreased lifespan [their average lifespan is 37 weeks (*Kondratov et al., 2006*; *Sun et al., 2006*)]. Possibly due to the pleiotropic effects of *Bmal1*, results of studies such as food entrainment behavior with $Bmal1^{-/-}$ mice have been controversial (*Fuller et al., 2008*; *Mistlberger et al., 2008*; *Fuller et al., 2009*; *Pendergast et al., 2009*; *Storch and Weitz, 2009*; *Mistlberger et al., 2009a, 2009b*; *Mieda and Sakurai, 2011*). To circumvent these problems, conditional knockout techniques utilizing Cre recombinase and *loxP* sequences have been applied widely in the nervous system (*Gavériaux-Ruff and Kieffer, 2007*; *Taniguchi et al., 2011*). However, until recently, Cre drivers that can efficiently recombine a floxed allele in the majority of SCN neurons have not been identified (*Mieda and Sakurai, 2011*; *Musiek et al., 2013* but see *Husse et al., 2011*).

In this study, we examined the ability of a forebrain-specific Cre driver to excise floxed alleles of *Bmal1*, a critical component of the molecular oscillator, specifically from brain regions including the SCN. The resulting knockout mice were devoid of BMAL1 expression in the forebrain/SCN but not in other peripheral tissues. Unlike *Bmal1* global knockout mice, *Bmal1* forebrain knockouts did not display detectable defects in reproduction, activity, body weight, or life span, thereby providing an ideal model to study circadian physiology in the absence of a central clock. These genetically engineered mice are better experimentally controlled than mice that receive SCN lesions, which is a classic method

to induce arrhythmic behavior in rodents (*Meijer and Rietveld, 1989*). With SCN lesions, precise removal of the SCN tissue varies from mouse to mouse and requires laborious confirmation of the excision. In addition, ablation of the SCN destroys neuronal networks and communication pathways, rendering the network responses and mechanisms incapacitated.

With forebrain-specific *Bmal1* knockout mice, we examined the impact of defunct brain clocks on the organization of peripheral oscillators as well as their response to environmental signals such as light and food. Cre-mediated excision of *Bmal1* in the forebrain/SCN confers complete arrhythmicity to circadian behavior. In contrast to the abolition of rhythms in the SCN, all peripheral tissues in these mice sustained circadian rhythmicity; however, they lacked phase coordination and normal amplitude in constant darkness. The synchrony and the oscillatory amplitude of peripheral rhythms in these mice were completely rescued by exposure to light dark cycles but not fully in time-restricted feeding experiments. The feeding cues appear to control the phase expression of the peripheral clocks in a tissue-specific way and in a distinct manner from light. These results demonstrate that an intact central clock plays an essential role in driving normal circadian behavior as well as synchronized and robust circadian programs in peripheral clocks and that light and feeding act on individual peripheral clocks differently in the absence of a functional oscillator in the SCN.

## Results

### Forebrain/SCN-specific deletion of *Bmal1*

*Bmal1* (also known as *Arntl*, *Mop3*) is one of the essential components of the molecular oscillator, unique in that the single knockout of *Bmal1* can confer arrhythmicity both at the behavioral and molecular levels (*Bunger et al., 2000*). To generate a forebrain-specific knockout of circadian rhythms, we crossed floxed *Bmal1* (*Bmal1*fx/fx) mice (*Johnson et al., 2014*) to *Camk2a::iCre*BAC mice (*CamiCre* or *Cre*) (*Casanova et al., 2001*) and produced *CamiCre+*; *Bmal1*fx/fx (that we denote as 'Brain' Knockout, BKO, defined as a forebrain-specific knockout) as well as *Bmal1*fx/fx (Fx/Fx), *CamiCre+* (Cre), and *CamiCre+*; *Bmal1*fx/+ (Het) control mice. *Camk2a::iCre*BAC mice were used because their Cre expression is faithful to the endogenous *Camk2a* gene expression pattern: it is neuron-specific, forebrain-enriched (*Casanova et al., 2001*), and expressed highly in the SCN as observed in Cre-dependent fluorescence reporter mice, *CamiCre+*; *tdTom+* (*Figure 1A–E*). Although there are more than 37 different *Camk2a::Cre* transgenic lines reported in Mouse Genome Informatics database (http://www.informatics.jax.org/searchtool/Search.do?query=Camk2a-cre&page=featureList), the majority do not appear to express well in the SCN (e.g., http://connectivity.brain-map.org/transgenic/experiment/81162458), likely because of shorter promoter regulatory sequences and position effects. The other advantage of this Cre driver is that its Cre expression is post-natal and starts at P3 (*Casanova et al., 2001*), thus reducing potential developmental effects. *Bmal1*fx/fx mice were generated by inserting *loxP* sites into the introns surrounding exon 4, which codes for the DNA-binding bHLH domain (*Johnson et al., 2014*). Upon Cre recombination, exon 4 is deleted in BKO brain tissues that express *Camk2a::iCre*. The predicted BMAL1 sequence encounters a STOP codon after a stretch of 18 non-native residues following native coding exon 3. The resulting protein is 91 amino acids long (vs 626 amino acids of native BMAL1) and is almost identical to the truncated BMAL1 protein of *Bmal1*−/− mice (*Bunger et al., 2000*); both proteins contain no known functional domains and do not accumulate to a detectable level.

In contrast to global *Bmal1* knockouts, the reproduction, longevity, and health of BKO mice were as robust as that of wild-type mice. The fertility of both males and females was indistinguishable from wild-type. At 2 months and 4–6 months of age, the body weights of BKO were found to be nearly identical to Fx/Fx, Cre, and Het mice (*Figure 1—figure supplement 1*).

In order to assess the deletion of *Bmal1* in the SCN, we examined the circadian mRNA expression pattern of *Bmal1* and *Per2*, a key *Bmal1* target gene, by in situ hybridization after 2 days of constant darkness (DD). At all seven time points taken over a period of 24 hr, BKOs showed a low level of *Bmal1* and *Per2* mRNA in the SCN as well as in the entire coronal brain section (*Figure 1F,G*). In more detailed analysis, control mice (Cre, Fx/Fx) showed intense mRNA expression of *Bmal1* in the SCN at its peak time (CT18) as well as circadian fluctuation (CT18 vs CT6) (*Figure 1F,H*; *Figure 1—figure supplement 2*). Accordingly, *Per2* mRNA was also robustly expressed in the SCN in a circadian manner (*Figure 1G,I*; *Figure 1—figure supplement 2*). Het mice showed reduction in the *Bmal1* mRNA abundance, yet cycling of *Per2* mRNA was retained in the SCN (*Figure 1—figure supplement 2*), indicating

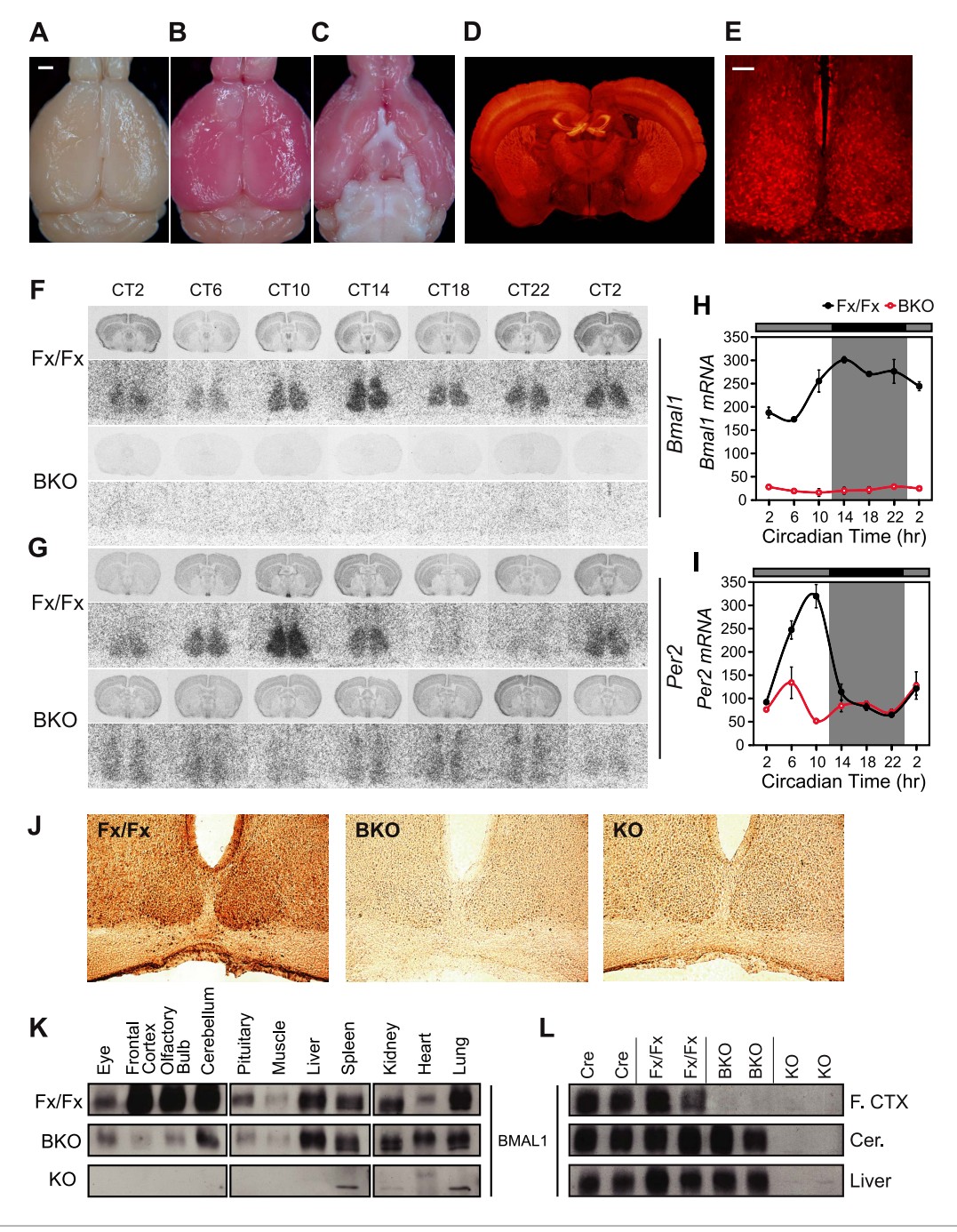

**Figure 1**. *CamiCre+; Bmal1*^fx/fx mice are forebrain/SCN-specific *Bmal1* knockouts. (**A**) Dorsal view of a *CamiCre*-; *tdTom*+ control brain. Bright field. Scale bar: 1 mm. (**B**) Dorsal view of a *CamiCre*+; *tdTom*+ brain. Bright field. (**C**) Ventral view of (**B**). (**D**) Fluorescence image of a coronal brain section containing the SCN from a *CamiCre*+; *tdTom*+ mouse. (**E**) Confocal image of the SCN showing tdTomato expression in the *CamiCre*+; *tdTom*+ mice. Scale bar: 50 μm. (**F**, **G**) Coronal brain sections at the level of the SCN, prepared from mice sacrificed every 4 hr starting at CT2 following 2 days of DD, were hybridized in situ to examine *Bmal1* (**F**) and *Per2* (**G**) mRNA levels in *CamiCre*+; *Bmal1*^fx/fx (BKO) mice (n = 3, except n = 4 for CT10, 14, 18) and *Bmal1*^fx/fx (Fx/Fx) control littermates (same as BKO, except n = 1 for CT18 *Bmal1* and n = 3 for CT18 *Per2*). Below each coronal brain section is a close-up of the SCN. (**H**, **I**) Optical density graphs of the in situ time course data displayed in (**F**) and (**G**). Shown are mean ± SEM, with significant effect of genotype in (**H**) [$F_{1,46}$ = 40.7, p < 0.0001] and (**I**) [$F_{1,44}$ = 1148, p < 0.0001] by GLM ANOVA. Tukey–Kramer multiple comparison post-tests (p ≤ 0.05) showed that in contrast to Fx/Fx littermates, neither *Bmal1*

*Figure 1. Continued on next page*

*Figure 1. Continued*

nor *Per2* mRNA is rhythmic in BKOs. (**J**) Immunohistochemistry for BMAL1 on SCN-containing coronal sections of Fx/Fx (n = 3), BKO (n = 3), and *Bmal1*$^{-/-}$ (KO, n = 1) mice sacrificed at ZT16. Captured with an 20× objective. (**K**, **L**) Western blot analysis of BMAL1 in BKOs sacrificed at ZT16. (**K**) Western blot of various tissues in Fx/Fx (n = 2), BKO littermate (n = 2), and KO (n = 1) mice. (**L**) Western blot of frontal cortex, cerebellum, and liver in Cre (n = 2), Fx/Fx (n = 2), BKO (n = 3), and KO (n = 2) mice. F. CTX: frontal cortex, Cer: cerebellum.

The following figure supplements are available for figure 1:

**Figure supplement 1**. Weights of *Bmal1* brain knockout mice are similar to controls at 2 months and 4–6 months of age.

**Figure supplement 2**. Additional in situ analysis of *Bmal1* brain knockout mice.

that 50% wild-type *Bmal1* mRNA is sufficient to sustain *Per2* expression in the SCN. In contrast, in BKO mice, *Bmal1* mRNA was reduced ~90% in the SCN (*Figure 1F,H*; *Figure 1—figure supplement 2*), and *Per2* mRNA cycling was significantly attenuated (~20% WT levels at CT10, see *Figure 1G,I*).

To confirm that no BMAL1 protein was expressed in the SCN of BKOs, we performed immunohistochemistry in the SCN at Zeitgeber Time (ZT) 16. In contrast to the intense BMAL1 staining in the SCN of Fx/Fx controls, BMAL1 staining in the BKO and global *Bmal1* knockout SCN was reduced to background levels (*Figure 1J*). To further characterize the tissue specificity of the BKOs, we used Western blot analyses to measure BMAL1 protein levels in various brain and body tissues at ZT16 (*Figure 1K,L*). In contrast to Fx/Fx and Cre controls, BMAL1 (~69 kD) was detected only faintly in BKO frontal cortex and olfactory bulb (*Figure 1K*), likely due to glial and non-neuronal expression of BMAL1 since *Camk2a* is not expressed in glial cells (*Lin et al., 1987*). Levels of BMAL1 in the cerebellum of BKOs were similar to Fx/Fx and Cre control mice (*Figure 1L*). Outside of the brain, like Fx/Fx mice but unlike global *Bmal1*$^{-/-}$ mice, BKOs showed equivalent BMAL1 expression levels in the liver, spleen, kidney, heart, and lung (*Figure 1K*). Taken together, these results confirm the forebrain specificity of our *Bmal1* conditional knockouts.

## Lack of circadian rhythms in BKOs

To assess whether forebrain/SCN *Bmal1* is necessary for normal circadian behavior, we examined circadian rhythms of locomotor activity in Cre, Fx/Fx, Het, and BKO mice in a 12 hr:12 hr light:dark (LD) cycle followed by DD and constant light (LL) conditions (*Figure 2*). In both DD and LL, control mice (Cre, Fx/Fx, and Het) expressed robust circadian rhythms of activity (*Figure 2A,C,D*). The locomotor activity of Het mice was indistinguishable from that of the other controls. In contrast to controls, *Bmal1* brain knockouts exhibited no circadian rhythm of activity in DD or in LL (*Figure 2B–D*). Out of 31 BKO mice, none had a period in the circadian (18–30 hr) range in DD. In LL, only 2 out of 21 mice had a period in the circadian range (24.1 and 20.9 hr). However, they showed extremely low amplitudes of circadian activity (0.025 and 0.019, respectively, fraction of power in the circadian range, where 1.0 represents the total power in the normalized power spectrum), and spectral analysis using Fast Fourier Transform (FFT) did not detect periodicity in the circadian range in any of the BKOs under either DD or LL conditions (*Figure 2D*).

Unexpectedly, total daily activity of BKOs was significantly higher than that of control mice both in DD and LL (*Figure 2E*). Previous studies have found that the locomotor activity of *Bmal1*$^{-/-}$ mice is decreased dramatically below wild-type levels (*Bunger et al., 2000*) and that transgenic BMAL1 expression in skeletal muscle rescues this diminished activity (*McDearmon et al., 2006*). Taken together, these results suggest that a peripheral defect in BMAL1 is responsible for the low activity levels that had been reported in global *Bmal1*$^{-/-}$ mice. In BKOs (which do not have this peripheral defect so that locomotion is normal), the absence of a central circadian gating mechanism, which normally suppresses activity during the subjective daytime (*Davis and Menaker, 1980*), likely contributes to an increase in overall levels of activity.

## Entrainment to light cycles and masking

One of the notable differences between BKO and SCN-lesioned mice is that BKOs display apparent entrainment to light/dark cycles (probably because BKOs are spared from surgical interruption of retinal

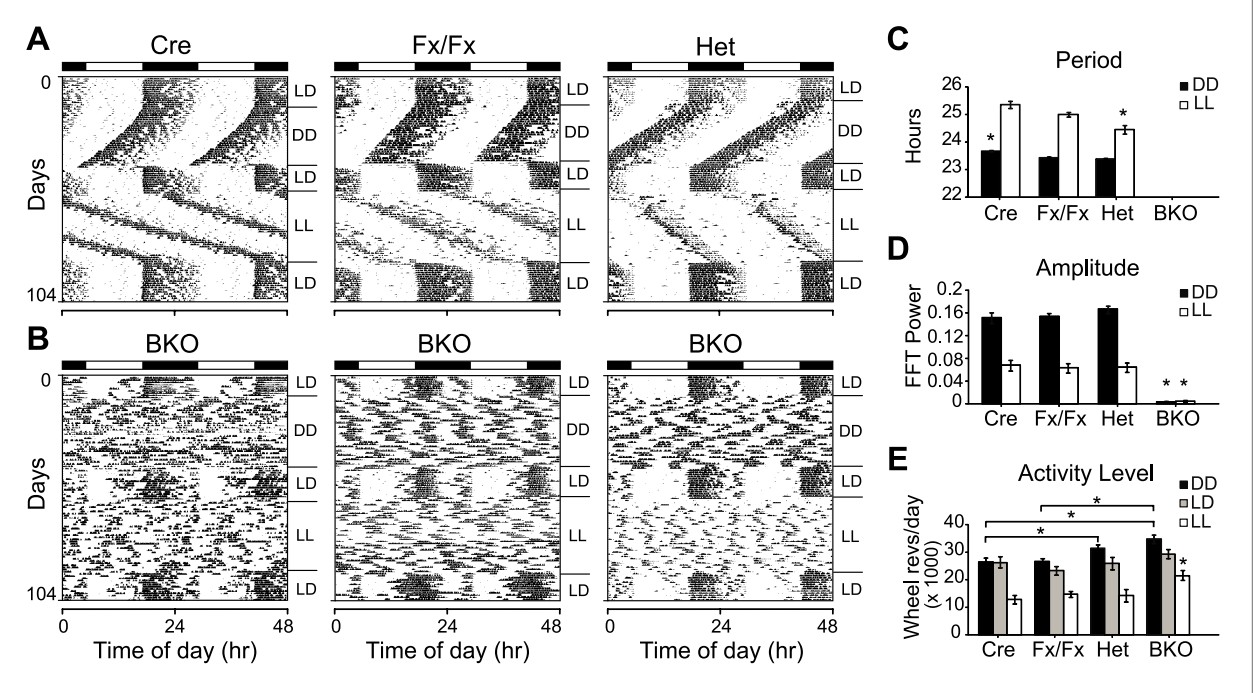

Figure 2. Complete loss of circadian rhythmicity of *Bmal1* brain knockout mice in constant darkness and constant light. (A, B) Representative actograms of daily wheel-running activity of Cre, Fx/Fx, Het, and BKO mice. Activity records were double plotted, with each day being represented beneath and also to the right of the preceding day. Horizontal black and white bars at the top of each actogram represent lights off and on, respectively. Mice were housed in LD, released into DD for 4 weeks, returned to LD for 2 weeks, released into LL for 4 weeks, and then returned to LD. GLM ANOVA and Tukey–Kramer multiple comparison post-tests were used to compare genotypes tested. BKOs showed no significant periodicity in DD or LL. (C–E) Period (C), amplitude of circadian rhythm (D), and activity levels (E) in Cre mice (n = 28 for DD, 12 for LD, 12 for LL), Fx/Fx mice (n = 31 for DD, 18 for LD, 18 for LL), Het mice (n = 31 for DD, 13 for LD, 15 for LL), and BKO mice (n = 31 for DD, 16 for LD, 21 for LL). Bar graphs show mean ± SEM. (C) Free-running period was determined using $\chi^2$ periodogram for days 1–28 of DD or LL. A significant effect of genotype on period was found for both DD [$F_{3,120}$ = 164,319, p < 0.0001] and LL [$F_{3,65}$ = 20,429, p < 0.0001]. In DD, Cre mice were found to have a slightly longer period than the other two control groups (*p ≤ 0.05). In LL, Het mice were found to have a shorter period than the other two control groups (*p ≤ 0.05). Nevertheless, all three control groups showed a free-running period in both DD and LL similar to that reported for WT mice. (D) DD and LL amplitude of circadian rhythm represented by FFT in the circadian range. A significant effect of genotype on amplitude was found for both DD [$F_{3,120}$ = 150, p < 0.0001] and LL [$F_{3,65}$ = 23.72, p < 0.0001], with BKO mice having significantly lower amplitude (*p ≤ 0.05) in both DD and LL. (E) Total daily DD, LD, and LL activity levels, in wheel revolutions per 24 hr. BKO activity levels were found to be significantly higher than those of controls in DD [$F_{3,119}$ = 9.00, p < 0.0001] (*p ≤ 0.05) and LL [$F_{3,65}$ = 5.84, p = 0.0014] (*p ≤ 0.05) but not in LD.

inputs to the SCN which usually occurs in lesion experiments) (*Figure 2*). However, the activity onset of BKOs in LD was variable and on average earlier than controls. As a result, daytime activity level was higher in BKOs (15.58 ± 1.8% of total daily activity) compared to the three control groups (3.81 ± 0.96%, average of the three control groups ± SEM) (*Figure 3—figure supplement 1*). Yet it was unclear whether the pre-dark activity of *Bmal1* brain knockouts around the light/dark transition was due to poor entrainment of a residual circadian pacemaker or to a defect in masking.

In order to test whether *Bmal1* brain knockouts entrain to light, we used a skeleton photoperiod (LDLD 1:10:1:12, *Figure 3A–E*). The activity patterns of Cre, Fx/Fx, and Het mice during the skeleton photoperiod were virtually indistinguishable from the activity patterns in LD 12:12 (*Figure 3A*). In contrast, none of the 10 BKO mice used in this study entrained normally to the skeleton photoperiod (*Figure 3B*): 3 BKO mice showed activity that resembled their activity in DD (*Figure 3B*, *left*), 2 BKO mice displayed a bimodal activity pattern (*Figure 3B*, *right*), and 5 BKO mice exhibited activity that was intermediate between these two phenotypes (*Figure 3B*, *middle*). Unlike control mice, BKO mice distributed their running activity equally between the subjective day and the subjective night of the skeleton photoperiod (*Figure 3C,D*). In fact, controls spent 97.9% of their activity during the skeleton night, compared to only 52.5% for BKO mice (*Figure 3E*). The amount of activity that BKOs spent during skeleton day vs night was not significantly different. This type of abnormal entrainment to skeleton photoperiods is

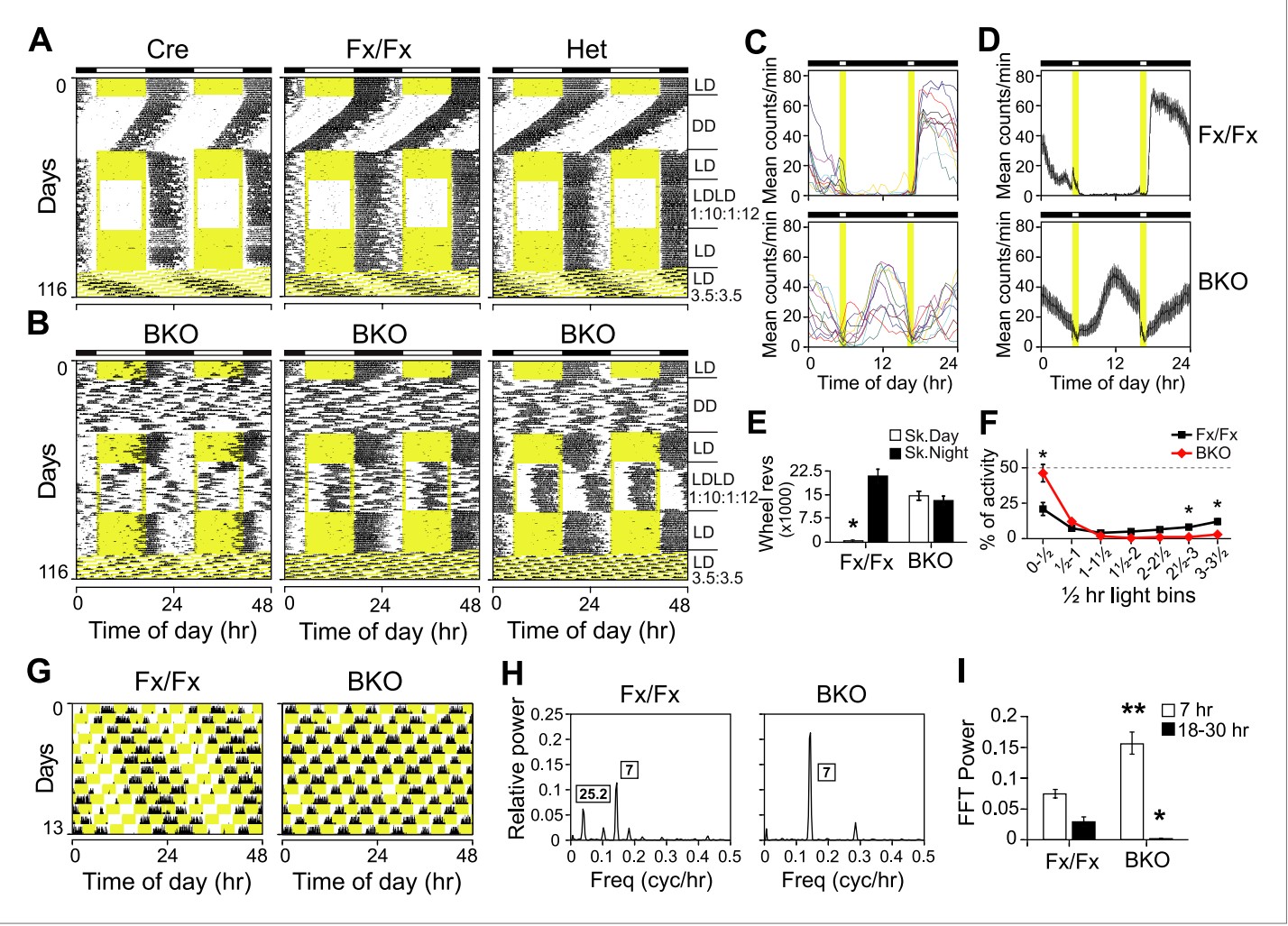

**Figure 3**. Both entrainment and masking are abnormal in *Bmal1* brain knockout mice. (**A**, **B**) Representative double-plotted actograms of daily wheel-running activity of Cre, Fx/Fx, Het, and BKO mice. Mice were housed in LD, released into DD for 4 weeks, returned to LD for 2 weeks, introduced to a skeleton photoperiod (LDLD 1:10:1:12) for 4 weeks, returned to LD for 2 weeks, and then switched to an ultradian cycle (LD 3.5:3.5) for 3 weeks. Shown are days 1–24 of the skeleton photoperiod and days 3–15 of LD 3.5:3.5. (**C**, **D**) Average activity profiles of Fx/Fx (n = 12) and BKO littermates (n = 10) during LDLD 1:10:1:12. (**C**) Average profiles for individual mice; (**D**) the ensemble average for the entire group. Each data point represents counts per minute averaged across a 6-min bin (±SEM for **D**). (**E**) Total activity counts (mean ± SEM) during 10 hr of subjective day (Sk. Day) and 10 hr of subjective night (Sk. Night) of LDLD 1:10:1:12. In contrast to Fx/Fx littermates, BKOs are equally active during skeleton day vs skeleton night. *p < 0.0001 by t-test. (**F**) Masking of BKO (n = 14) and Fx/Fx littermates (n = 9) during specific 0.5 hr light bins of LD 3.5:3.5. Total activity (mean ± SEM) during each 30 min light bin is divided by the total of that bin plus the corresponding dark bin. Thus the first data point for each genotype represents the distribution of activity between the first 30 min of light and the first 30 min of dark. There was a significant interaction between the genotype and time using Repeated Measures GLM ANOVA [$F_{1,160}$ = 22.3, p < 0.0001]. Specifically, days 1, 6, and 7 were found to be significantly different by Tukey–Kramer multiple comparison post-tests (*p ≤ 0.05). (**G**) Representative double-plotted actograms of daily wheel-running activity of Fx/Fx (*left*) and BKO (*right*) littermates. These are the same mice as those shown in (**A**, middle panel) and (**B**, middle panel), respectively. Light phases are indicated in yellow to show the structure of the LD 3.5:3.5 cycle as well as to help visualize the occurrence of wheel-running activity under this schedule. Note that, after 1 week of this cycle, initial phase relationships are regained. (**H**) Representative amplitude power spectra of Fx/Fx and BKO mice, from FFT analyses performed on the same activity records shown in (**G**). The highest peaks for both genotypes were in the 7-hr range, corresponding to the LD 3.5:3.5 schedule. The second-highest peak for the controls was in the circadian (18–30 hr) range (*left*). None of the BKOs was found to have a period in the circadian range (*right*). (**I**) Group averages of amplitudes in the 7-hr range and in the circadian range for the same Fx/Fx and BKO mice as in (**F**). Compared to controls, BKOs had significantly higher amplitude in the 7-hr range (**p < 0.0001 by t-test) and significantly lower amplitude in the circadian range (*p < 0.0060 by t-test).

The following figure supplement is available for figure 3:

**Figure supplement 1**. Masking in *Bmal1* brain knockout mice.

reminiscent of the entrainment behavior of pinealectomized sparrows, which behave as a population of circadian oscillators that entrain with two opposite phase relationships to the skeleton photoperiod (*Takahashi and Menaker, 1982*). Therefore, we conclude that entrainment in *Bmal1* brain knockout mice is abnormal under skeleton photoperiods, consistent with the idea that the coherence of phase during entrainment may be compromised. Overall this suggests that the periodicity seen in LD 12:12 in BKOs arises from entrainment of a population of driven oscillators in which phase control is labile.

Since the skeleton photoperiod and LD photoperiod results both indicated some impairment of masking in BKOs, we next tested whether negative masking (the ability of light to acutely suppress locomotor activity) in *Bmal1* brain knockouts was defective. To assess masking in BKO mice, we used an LD 3.5:3.5-hr light–dark cycle (*Redlin et al., 2005*) (*Figure 3B,F,G*). Mice are unable to entrain to LD 3.5:3.5 cycles, thus permitting the masking and the entraining effects of light to be distinguished. Maintaining this schedule for a week ensures the testing of all phases of a circadian cycle (*Figure 3G*). On average, BKOs exhibited significantly less activity during the light portions of LD 3.5:3.5 (4.68 ± 0.71% of total activity vs 9.56 ± 1.83% for Fx/Fx littermates, p < 0.032 by t-test). Fx/Fx mice display more activity during the light portions of LD 3.5:3.5 because the circadian clock is promoting activity despite the presence of light (*Figure 3G*, *left*). While Fx/Fx mice have an underlying circadian component of activity under LD 3.5:3.5, BKOs have none (*Figure 3G–I*; *Figure 3—figure supplement 1*). Aside from this difference, when we examined the time course of the masking response, it became apparent that the masking of BKOs is delayed compared to controls (*Figure 3F*; a score of 50% indicates failure to mask). Thus BKOs failed to mask during the first 30 min of the light phases. Plotting actograms on a 7-hr time scale aids in visualization of this masking impairment in BKOs (*Figure 3—figure supplement 1*). These entrainment and masking studies led us to conclude that BKOs lack an endogenous clock but also exhibit some impaired masking.

## Phase synchrony and desynchrony in peripheral tissues of BKOs

To analyze the temporal organization of peripheral oscillators at an organismal level, we crossed the BKOs to PER2::LUC mice (*Yoo et al., 2004*) and monitored the bioluminescence signals from the SCN and other tissues in the body. Compared with the robust oscillation of the control SCN, the SCN of BKOs exhibited blunted expression patterns with low-amplitude but detectable rhythms, which were close to 24 hr (23.54 ± 0.29 hr in Fx/Fx control SCN vs 22.99 ± 2.36 hr in BKO SCN for mean period length ± SD, *Figure 4A*). These rhythms are likely to be derived from glial cells and/or residual neurons in which Cre-mediated excision of floxed *Bmal1* was incomplete (see *Figure 1J*) or possibly from residual stochastic oscillations that we have observed in the SCN of global *Bmal1* knockouts (*Ko et al., 2010*). Rhythmic expression in the dorsomedial hypothalamus (DMH) in BKOs was also significantly attenuated. In contrast to the SCN and DMH, peripheral tissues retained persistent circadian rhythms (*Figure 4A*), consistent with the normal expression of BMAL1 in peripheral tissues (*Figure 1K*).

These observations prompted us to use the forebrain-specific knockout mice to re-investigate the relationship between the brain's central pacemaker and circadian oscillations of peripheral tissues. Previous studies have addressed a similar question by lesioning the SCN (*Yoo et al., 2004*; *Tahara et al., 2012*; *Saini et al., 2013*). Unlike SCN-lesioned mice, however, BKOs preserve intact neural networks and show normal or enhanced levels of behavioral activity in LD and in constant conditions. This new analysis allows us to determine whether deleting BMAL1 (thus inactivating the molecular clock) in the forebrain, without destroying the structure of the SCN, is sufficient to affect the organization of peripheral rhythms.

Age-matched pairs of mice (Fx/Fx littermate control and BKO) were maintained in DD for more than 30 days (30–44 days) before harvesting eight different tissues for real-time reporting of circadian gene expression. As a light synchronized control group, tissues were also collected from mice in LD 12:12. Consistent with previous studies (*Yoo et al., 2004*; *Tahara et al., 2012*), mean period length was variable yet characteristic from tissue to tissue (*Figure 4B,C*). The difference of the mean periods between the Fx/Fx control and BKO mice, however, was not significant either in LD or DD, except for the SCN and DMH (*Figure 4B,C*).

To analyze the temporal organization of the rhythms of each tissue, we constructed a phase map by plotting the peak on the second day in the luminometry recording. Fx/Fx control mice displayed tightly clustered phases in peripheral rhythms in both LD and DD (*Figure 5A,B,E*). On the other hand, while BKOs exhibited relatively coherent rhythms in peripheral tissues in LD (*Figure 5C,E*; *Figure 5—source data 1A*), their phases were significantly dispersed among animals in DD (*Figure 5D,E*; *Figure 5—source data 1A*). In order to measure the degree of phase coherence of peripheral clocks

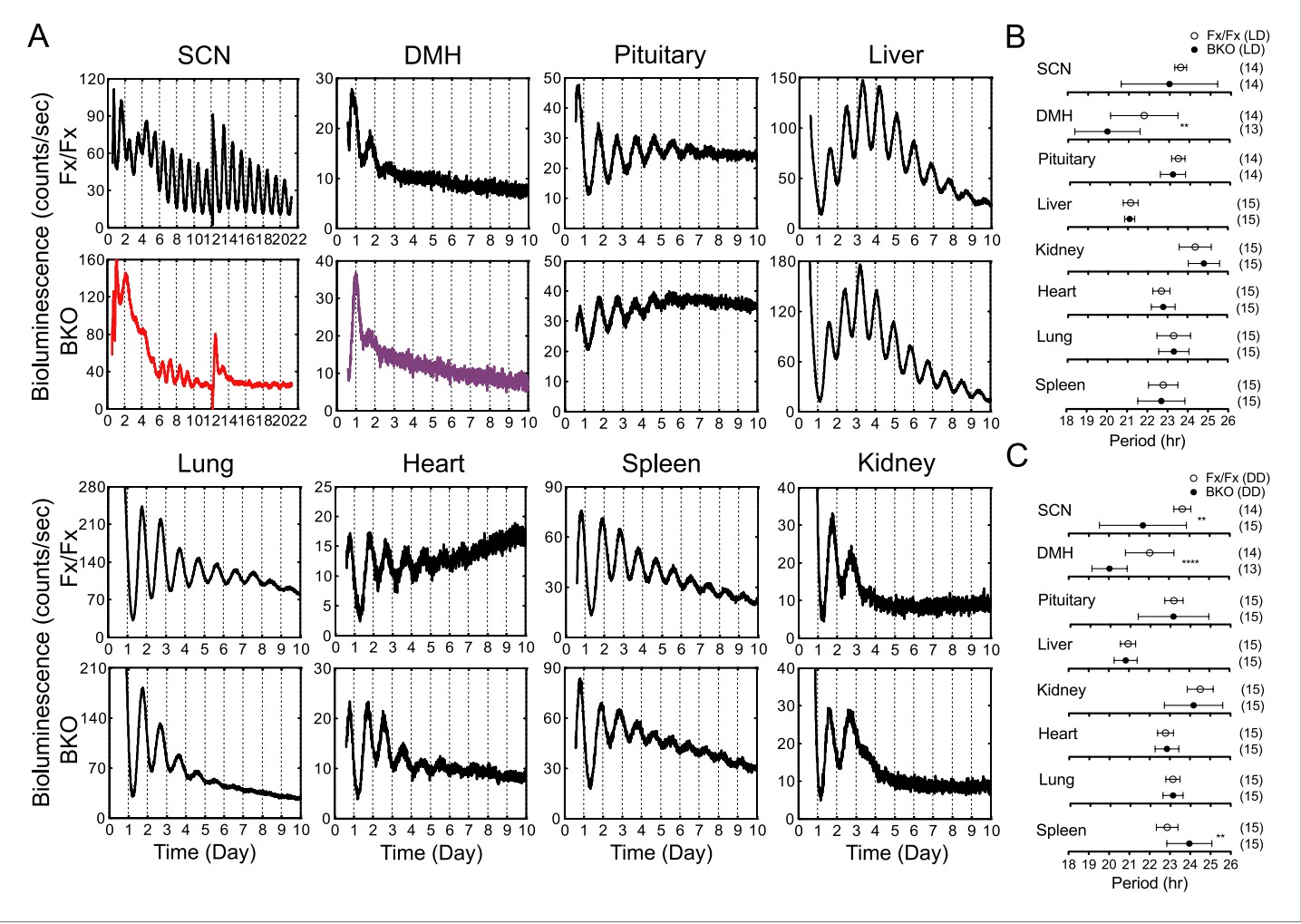

**Figure 4**. Real-time reporting of circadian expression of PER2::LUC in the forebrain/SCN knockout mice. (**A**) Representative records of bioluminescence reporting of circadian expression from various tissues in Fx/Fx and BKO mice. Tissues were prepared from mice in LD. PMT counts are plotted against the LD cycle of day 1. Shown are 10 days of continuous recording after explant preparation, except for the SCN for which medium was changed on day 12. (**B**, **C**) Period plots of various tissues harvested from Fx/Fx control (open circle) and BKO (dark circle) mice in LD (**B**) and DD (**C**). The sample size is indicated on right. Shown are mean period ± SD. **p < 0.01, ****p < 0.0001 by t-test.

within individual animals, the circular variance of peak bioluminescence of all tissues in each mouse was compared (*Figure 5—source data 1B,C*). Statistical analysis (*Figure 5F,G*) showed that BKO cultures from the DD condition have a significantly wider phase distribution among organs (*Figure 5G*, p = 0.0008 by Mann–Whitney test). This result indicates that a loss of phase coordination occurred in peripheral clocks within individual BKO animals when placed in DD. In contrast, we found that BKOs displayed well-phased rhythms from tissue to tissue when maintained in LD (*Figure 5F*, N.S. by Mann–Whitney test), suggesting that the internal phase synchrony is still maintained even in the absence of a functional master pacemaker when the animals are exposed to LD cycles.

## Decreased oscillatory amplitude in peripheral tissues of BKOs

In addition to period and phase, we also estimated amplitude because there was a trend for oscillatory amplitudes in peripheral tissues from BKOs to be lower than that of controls (*Figure 6—figure supplement 1*). To compare across different experiments, all time-series data were adjusted for PMT background counts and PMT gain, before applying FFT-NLLS computation as developed previously (*Izumo et al., 2006*). Only normalized amplitudes were compared in this study. For the SCN and DMH in which *Bmal1* is deleted from the majority of neurons, only residual amplitude was detected

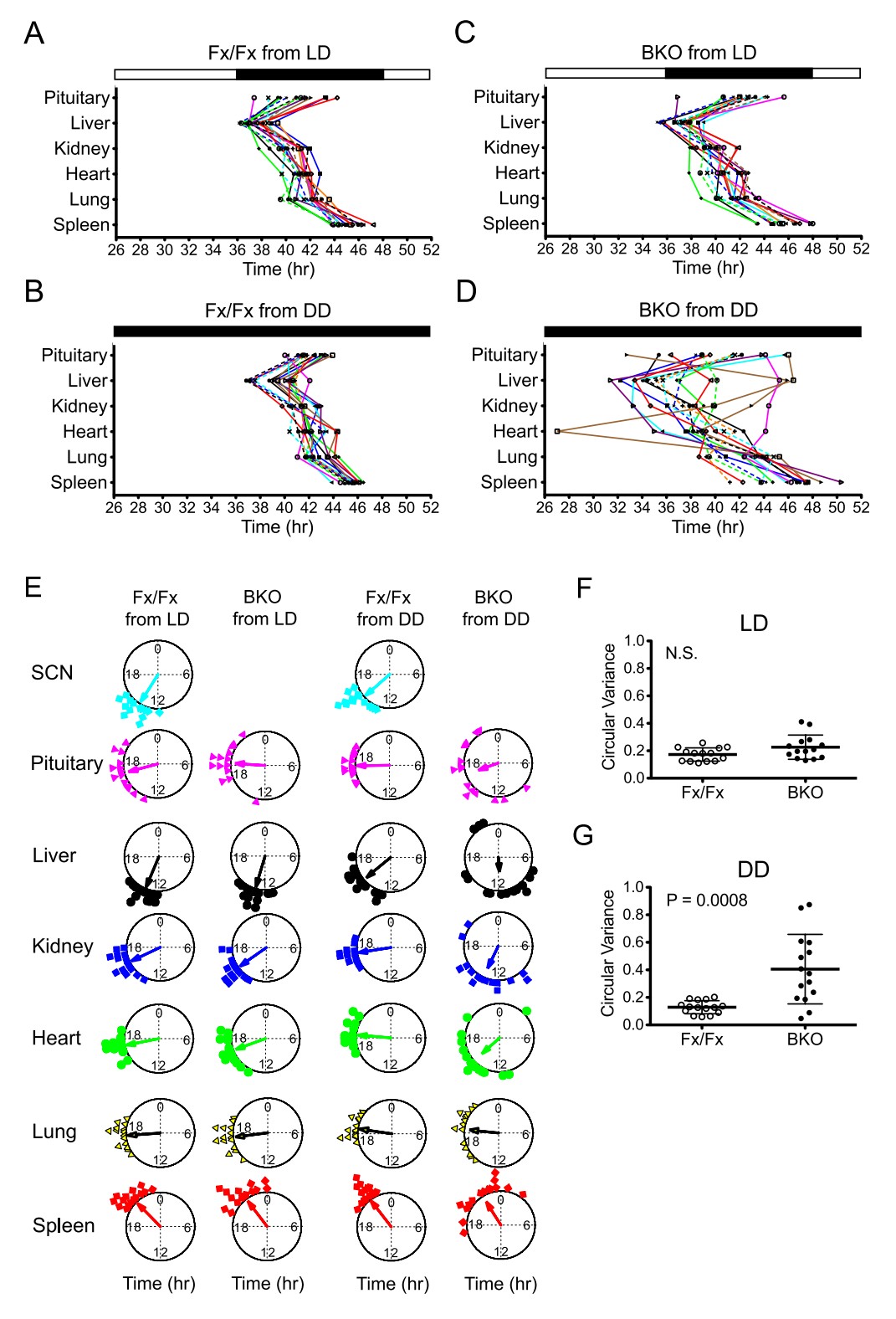

**Figure 5**. Phase analysis of circadian expression of various peripheral tissues from the forebrain/SCN knockout mice. (**A**) Phase map of circadian rhythms of various peripheral tissues from Fx/Fx control mice in LD (n = 15, except for pituitary n = 14). The peak phases (or averaged peak phases where more than two pieces were prepared from the same tissue) in the second cycle were plotted against the LD cycle of the day when explant cultures were prepared. *Figure 5. Continued on next page*

*Figure 5. Continued*

Tissues from the same animal are connected by colored lines with matched symbols. (**B**) Phase map of Fx/Fx control mice in DD (n = 15). The peak phases (or averaged peak phases) in the second cycle were plotted against the predicted onset of activity (=CT12) of the day when the explant cultures were prepared. (**C**) Phase map of BKOs in LD (n = 15, except for pituitary n = 14). (**D**) Phase map of BKOs in DD (n = 15). The phases were mapped against the predicted onset of activity (= CT12) of paired Fx/Fx control mice. (**E**) Circular plots of peak bioluminescence rhythms in the SCN and peripheral tissues presented in (**A–D**). A circle represents a 24-hr clock, and peak phases of bioluminescence rhythms of individual tissues were calculated as angles and plotted as colored symbols outside the circle. Each tissue is denoted by the same color and symbol scheme. The direction of the arrow indicates mean phase angle, with the length of the arrow expressing the strength of phase clustering. The sample size, mean phase angle, and circular variance for each dataset are summarized in *Figure 5—source data 1A*. (**F**, **G**) Scattered plot of circular variance in each individual mouse. Degree of variance among the peak phase values of pituitary, liver, kidney, heart, lung, and spleen in each individual mouse was calculated and expressed as circular variance (see *Figure 5—source data 1B,C*) and compared between Fx/Fx controls and BKOs in LD (**F**) and DD (**G**). The bar is ±SD. A significant effect was found between Fx/Fx control and BKO mice in DD (p = 0.0008 by Mann–Whitney test) but not in LD (p = 0.0627 by Mann–Whitney test). N.S. = not significant.

The following source data is available for figure 5:

**Source data 1**. (**A**) Summary of statistical analysis of circular plots presented in *Figure 5*. (**B**) Statistical analysis of circular variance for peak bioluminescence in individual mice in LD. (**C**) Statistical analysis of circular variance for peak bioluminescence in individual mice in DD.

(*Figure 6A,B*). Likewise, the amplitude of the pituitary was lower in BKOs from LD, which was likely due to the partial deletion of BMAL1 (see *Figure 1K*). This decrease was further pronounced when BKO pituitary was harvested from DD (*Figure 6A,B*). For all other peripheral tissues, the relative amplitude was significantly decreased when a light-driven signal was removed from BKOs in DD, but the severe reduction was not observed in BKOs from LD (*Figure 6A,B*, except for liver). These damped rhythms, however, regained oscillation after receiving a stimulatory medium change (*Figure 6—figure supplement 1*), which suggests the possibility that phase desynchrony is taking place within individual tissues of BKOs.

To distinguish whether the decreased amplitude of BKO tissues is due to phase desynchrony between the cells or to lowered amplitude within the cells, we performed bioluminescence imaging on a peripheral BKO tissue from DD. Heart tissue was chosen for this experiment, because the heart showed both significant phase dispersion and severe amplitude reduction in DD (*Figure 5D*; *Figure 6B,C*). In this imaging, bioluminescence signals from heart tissue were insufficient to identify single cells in the field of view. Therefore, we developed a grid method to allow extraction of time-series data in a narrow area of the target-imaging sample (*Figure 6D*). The size of the grid was narrowed down to 40 μm × 40 μm. Circadian rhythms in the brightest 200 grids were aligned in the order of expression intensity in a heat map (*Figure 6E*). Consistent with the organotypic recording by LumiCycle luminometry (*Figure 6C*), the heart cells from a wild-type mouse exhibited robust oscillation, while those from a BKO displayed damped rhythms with broader phase distribution. The damped rhythms were also clear in a linear graph in which the top 50 time-series were plotted (*Figure 6F,G*). Amplitude analysis of these 50 time-series data showed that the oscillatory amplitude was significantly reduced within the grids in the BKO heart sample (*Figure 6H*). Furthermore, their phases were significantly dispersed (*Figure 6I*). These results suggest that the amplitude decrease in the BKO tissue is caused by a combined effect of amplitude reduction and phase desynchrony at the cellular level.

## Intact food-anticipatory activity in brain *Bmal1* knockouts

Because nutrient signals can provide a potent entraining cue for peripheral circadian oscillators (*Green et al., 2008*; *Bass and Takahashi, 2010*; *Mohawk et al., 2012*), we investigated whether scheduled food restriction (FR) could restore circadian temporal organization of peripheral oscillators in forebrain-specific *Bmal1* knockouts. First, to test the effect of forebrain *Bmal1* on food-anticipatory activity (FAA), we subjected BKOs to a schedule of FR under both LD and DD conditions (*Figure 7A*).

Both qualitatively and quantitatively, mice lacking forebrain *Bmal1* exhibited clear food-anticipatory behavior in LD that was similar to that of control mice (*Figure 7B–E*). Notably, the mean FAA profiles of BKOs were enhanced in DD during temporal food restriction (*Figure 7C,G*), and the total daily

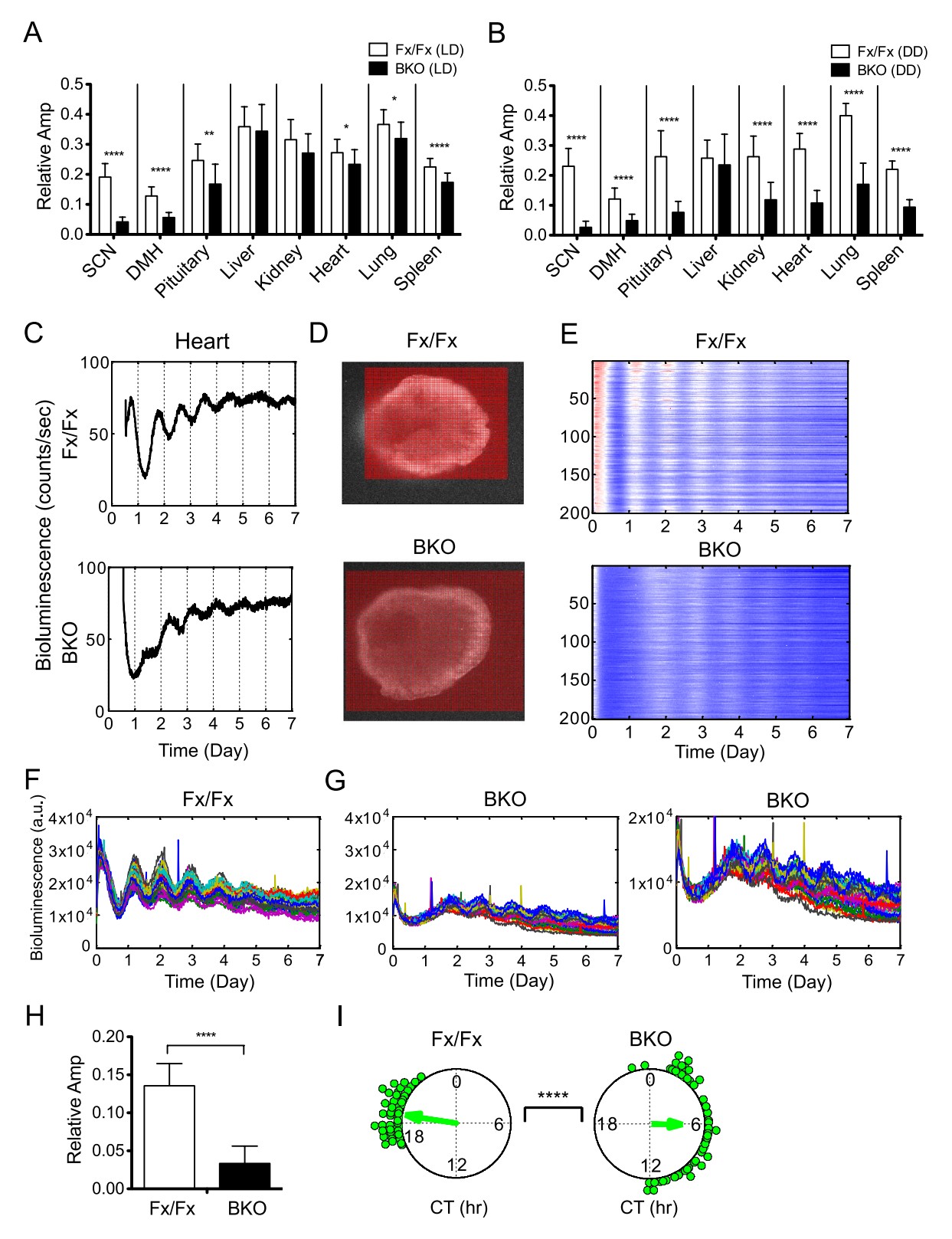

**Figure 6**. Decreased amplitude in peripheral tissues of BKOs. (**A**, **B**) Normalized amplitude of bioluminescence rhythms from Fx/Fx control (open bar) vs BKO (dark bar) in LD (**A**) and DD (**B**). Mean relative amplitude ±SD are shown. The sample size is the same as in *Figure 4B,C*. *p < 0.05, **p < 0.01, ****p < 0.0001 by t-test. (**C**) Representative records of real-time monitoring of circadian expression of heart tissues in Fx/Fx control (*upper*) and BKO

*Figure 6. Continued on next page*

*Figure 6. Continued*

(*lower*) mice maintained in DD. (**D**) Representative frames of bioluminescence imaging with grids over each heart tissue. (**E**) Heat maps of the brightest 200 time-series data beginning with the strongest signals in the grids shown in (**D**). (**F**, **G**) Linear traces of the top 50 of time-series data from the Fx/Fx (**F**) and BKO (**G**) heart tissue shown in (**E**). The Y-axis was expanded for the BKO sample in the most right graph. (**H**) Normalized amplitude quantified from the top 50 of time-series data shown in (**F**) and (**G**). ****p < 0.0001 by t-test. (**I**) Circular plots of peak phase values of the top 50 time-series data shown in (**F**) for Fx/Fx and (**G**) for BKO tissues. Degree of variance was compared between the two samples by bootstrapping simulation (****p < 0.0001).

The following figure supplement is available for figure 6:

**Figure supplement 1**. Real-time monitoring of circadian expression in forebrain/SCN knockout mice from DD.

activity levels in BKOs during FR were significantly higher (almost two-fold higher) than those of controls, despite the observation that BKOs exhibited a trend to eat less during FR (*Figure 7—figure supplement 1*). Surprisingly, FAA profiles showed that, during FR in DD, BKOs had extended activity for up to 15 hr before food availability, while controls allocated most of their activity to the 6-hr window of FAA (*Figure 7F,G*). Another study has found similarly longer FAA in *Bmal1*-deficient mice (*Takasu et al., 2012*). To further characterize the food-anticipatory behavior, we measured the time course over which FAA emerged in BKOs and controls after the start of food restriction. We found FAA development to be similar in the two genotypes under both LD and DD conditions. FAA appeared after about 2 days and reached a plateau after about 4 days in both genotypes (*Figure 7D,H*). The mean time by which FAA anticipated the daily onset of food (phase angle with respect to food presentation) was not significantly different between the two genotypes during FR in LD (*Figure 7E*). During FR in DD, the random activity in BKOs partially obscured their FAA; however, FAA was still recognizable above the baseline both in individual records (*Figure 7F*) and in the population activity profile (*Figure 7G*). This activity change was statistically significant (*Figure 7I*). Therefore, behavioral responses to temporal food restriction in BKOs were enhanced in DD.

Previously, the DMH was thought to be a possible candidate site of a food-entrainable oscillator (FEO) that regulates feeding behavior [(*Gooley et al., 2006*; *Mieda et al., 2006*) but see (*Landry et al., 2006*, *2007*; *Moriya et al., 2009*; *Acosta-Galvan et al., 2011*; *Landry et al., 2011*)]. To examine *Bmal1* deletion and *Per2* gene expression at this anatomical site, we conducted in situ analysis on the DMH. We collected brains at CT6 and CT18 on the third day of *ad lib* feeding in DD. Controls showed high expression levels of *Bmal1* mRNA in the DMH, albeit with no circadian fluctuation. On the other hand, BKOs had no significant *Bmal1* mRNA in the DMH at either time point (*Figure 7—figure supplement 1*). Despite a complete lack of *Bmal1* in the DMH, BKOs exhibited normal FAA, indicating that *Bmal1* in the DMH is not necessary for food entrainment. Surprisingly, BKOs showed *Per2* expression levels comparable to those of controls (*Figure 7—figure supplement 1*). Both genotypes displayed intense staining in the ventromedial portion of the compact nucleus of the DMH (DMC, *Figure 7—figure supplement 1*), and the difference between peak and trough *Per2* mRNA was not significant for either genotype. This result implies that transcriptional regulation of *Per2* in the DMH involves activation pathways (e.g., cAMP response element (CRE) and serum response element (SRE) mediated pathways) in addition to CLOCK:BMAL1 as seen previously (*Travnickova-Bendova et al., 2002*; *Gerber et al., 2013*).

## Restricted feeding selectively resynchronizes peripheral tissues

It has been shown that rhythms of peripheral tissues can be dissociated from the control of the central clock in the SCN and be re-entrained to a new phase by temporal feeding restriction (*Damiola et al., 2000*; *Stokkan et al., 2001*). The ability of the peripheral tissues to respond to feeding signals was further demonstrated in vivo (*Tahara et al., 2012*; *Saini et al., 2013*), with an observation that phase changes take place faster in the liver when the SCN is surgically lesioned. To assess the impact of feeding cues on the synchrony of peripheral rhythms, we subjected BKO; PER2::LUC mice to a restricted feeding schedule in the absence of light signals (*Figure 8A*). The duration of constant darkness was the same as the desynchronization experiment (*Figure 5*). During these >30 days, mice were fed *ad libitum* for the first 2 weeks, and after 5 days of a food ramp, temporal food restriction was performed for 2 weeks before harvesting the tissues for real-time gene expression recording.

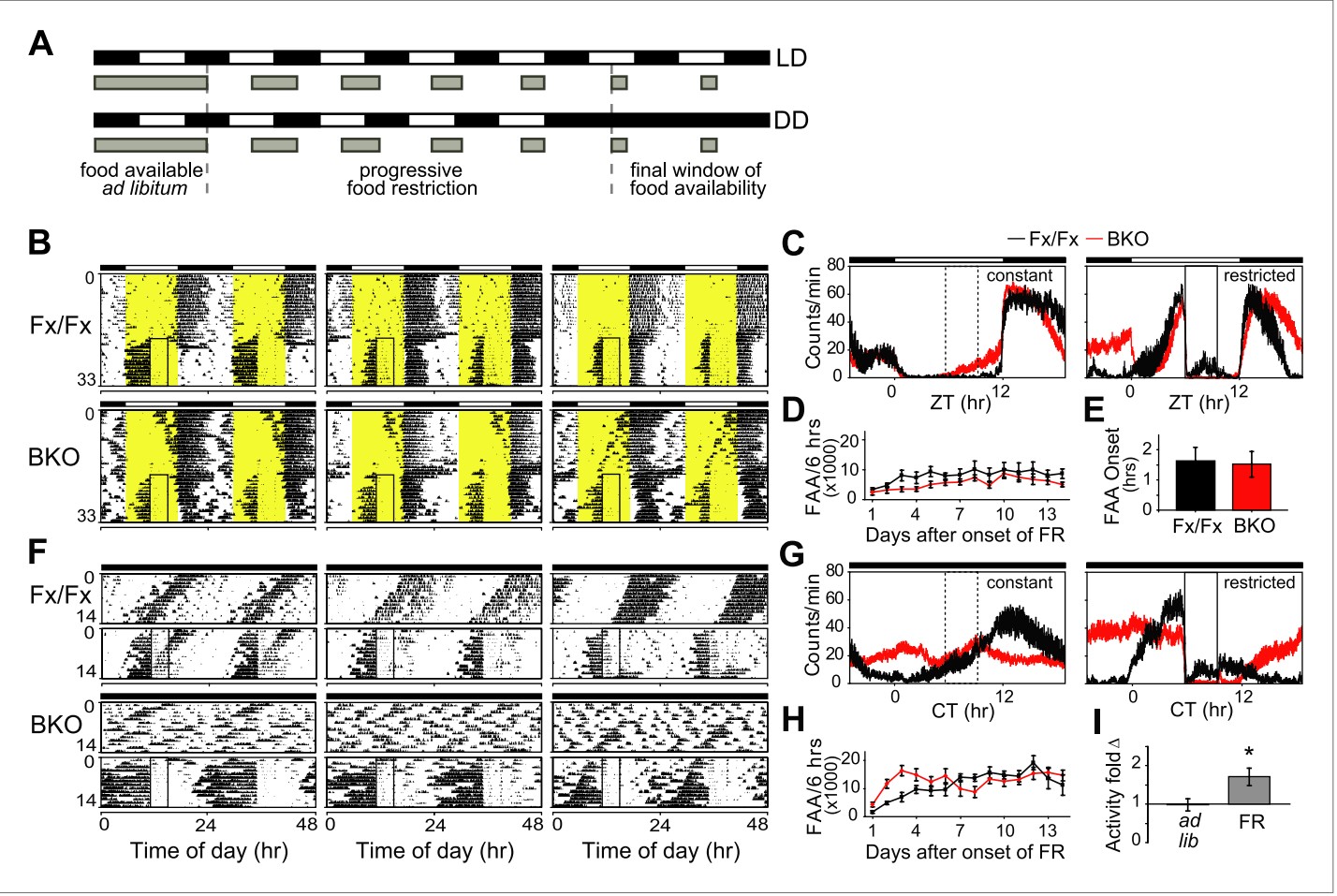

**Figure 7**. Intact FAA in *Bmal1* brain knockout mice. (**A**) Schedule of a gradual temporal FR protocol in LD (*upper*) or DD (*lower*). Horizontal black and white bars represent lights off and on, respectively. Gray bars represent food availability. Since abrupt shifts to FR can have a high morbidity in mice, a gentle temporal FR paradigm was used (FR ramp), decreasing the duration of daily food availability from constant to a final 4 hr window over the course of 5 days. Because free-running in DD can obscure FAA in control mice, we obtained DD baseline activity and then switched mice to 14 days of LD (not shown). We then transferred our mice directly from LD into DD FR following an LD food ramp (*lower*). It should be noted that once gradual FR began, the time of onset of food availability was the same each day. (**B**, **F**) Representative double-plotted actograms of daily wheel-running activity of 3 Fx/Fx control littermates and 3 BKO mice during *ad lib* feeding and under subsequent FR during LD (**B**) or DD (**F**). The boxed area toward the left side of each actogram indicates the daily interval of food availability under FR and yellow areas indicate time of lights-on during LD 12:12. After 5 days of gradually decreasing food availability, the final food availability window was ZT/CT6–10. For clarity, the 5-day FR ramp is not included in the boxed area (**B**). (**C**, **G**) Mean locomotor activity profiles of Fx/Fx littermates (n = 7) and BKO mice (n = 14) under *ad lib* feeding (*left*) and during FR days 8–14 (*right*) under LD (**C**) or DD (**G**). Each data point represents counts per minute averaged for each genotype across a 6-min bin (±SEM). The dashed boxed area (*left*) indicates, for comparison, the daily interval corresponding to subsequent food availability. The solid boxed area (*right*) indicates the daily interval of food availability under FR. (**D**, **H**) Time course of the development of FAA in Fx/Fx controls (n = 7) and BKO mice (n = 14) during LD FR (**D**) or DD FR (**H**). FAA is plotted as the total number of activity counts (mean ± SEM) allocated to a 6-hr time interval prior to mealtime, ZT/CT0–6 (**D**, **H**). (**E**) Number of hours by which FAA preceded daily meal times in Fx/Fx (n = 7) and BKO mice (n = 14) during LD FR. Wheel-running activity profiles were averaged for each individual during stable FR (as in **C**), and the average time of onset of FAA was determined as the time before food availability at which FAA rose to its half-maximum value (mean ± SEM). (**I**) Quantification of FAA under DD conditions in BKO mice (n = 14). Plotted is fold-change of wheel-running activity (counts per minute, mean ± SEM) in each mouse for CT0–6 (window of FAA) compared with CT10–24 (the rest of the day except for the window of food availability). During FR, increased locomotor activity during CT0–6 compared with CT10–24 was highly significant (*p = 0.0001 by paired t-test).

The following figure supplement is available for figure 7:

**Figure supplement 1**. Lack of *Bmal1* expression in the DMH of *Bmal1* brain knockout mice.

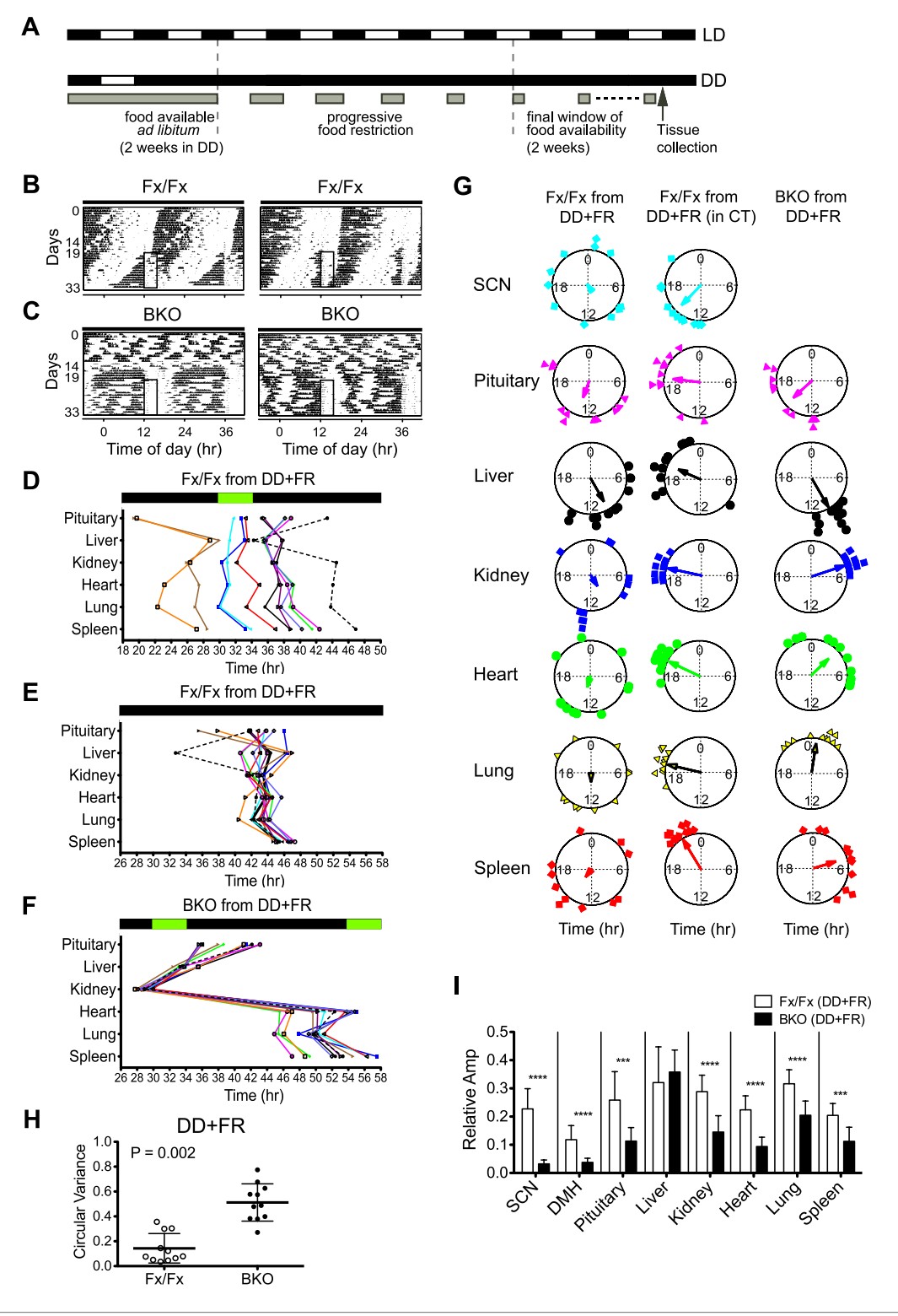

**Figure 8**. Effects of restricted feeding on circadian rhythms of peripheral tissues. (**A**) Schedule of a temporal FR protocol in DD for real-time circadian reporting assay. Horizontal black and white bars representing lights off and on for LD 12:12 are shown above as a reference for a daily feeding schedule. Gray bars represent food availability. After entrainment to LD 12:12, mice were released into and maintained in DD for a total of >30 days. In DD, mice were fed *ad libitum* for the initial 2 weeks and underwent a 5-day FR ramp of gradually decreasing food availability

*Figure 8. Continued on next page*

*Figure 8. Continued*

to reach the final food availability window of 4 hr (from ZT/CT6–10) for 2 weeks of FR. Tissues were harvested at 2 hr after removal of the food. (**B**, **C**) Representative double-plotted actograms of 2 Fx/Fx control (**B**) and 2 BKO mice (**C**) during 2 weeks of *ad lib* feeding followed by the 5-day FR ramp and subsequent 2 weeks of FR in DD. The boxed area toward the left side of each actogram indicates the daily interval of food availability under FR (ZT/CT6–10). (**D**) Phase map of circadian rhythms of various peripheral tissues from Fx/Fx controls in DD + FR (n = 11). The peak phases (or averaged peak phases) were plotted against the feeding time (green bar). Tissues from the same animal are connected by colored lines with matched symbols. (**E**) Phase map of (**D**) was converted to CT for the harvest time. (**F**) Phase map of BKOs in DD + FR (n = 11). The peak phases (or averaged peak phases) were plotted against the feeding time (green bars). (**G**) Circular plots of peak bioluminescence rhythms in the SCN and peripheral tissues presented in (**D**–**F**). The mapping strategy is the same as in *Figure 5E*. The sample size, mean phase angle, and circular variance for each dataset are summarized in *Figure 8—source data 1A*. (**H**) Scattered plot of circular variance in each individual mouse. Degree of variance among the peak phase values of pituitary, liver, kidney, heart, lung, and spleen in an individual mouse was calculated and expressed as circular variance (see *Figure 8—source data 1B*) and compared between Fx/Fx controls and BKOs in DD + FR. The bar is ±SD. Note that the degree of variance is not changed by phase data conversion from ZT to CT. A significant effect was found between Fx/Fx control and BKO mice (p = 0.0002 by Mann–Whitney test). (**I**) Normalized amplitude of bioluminescence rhythms from Fx/Fx control (open bar) vs BKO (dark bar) in DD + FR. Mean relative amplitude ±SD are shown. The sample size is the same as in (**D**, **F**). ***p < 0.001, ****p < 0.0001 by t-test.

The following source data is available for figure 8:

**Source data 1**. (**A**) Summary of statistical analysis of circular plots presented in *Figure 8*. (**B**) Statistical analysis of circular variance for peak bioluminescence in individual mice under restriction feeding.

Consistent with a previous report (*Storch and Weitz, 2009*), Fx/Fx control mice exhibited both free-running activity and FAA in constant darkness (*Figure 8B*). The peripheral tissues in these mice displayed variable phase relationships with respect to the feeding time (*Figure 8D*). However, when the peak phases were converted to the circadian time of the animal at harvest, the phases in a majority of tissues examined (except for pituitary and liver) became markedly clustered (*Figure 8E,G*; *Figure 8—source data 1A*), suggesting that the circadian rhythms in many peripheral tissues in wild-type mice are still dictated by the signals from the intact central clock. On the other hand, the peripheral tissues of BKOs showed distinct changes in phase expression patterns in response to restricted feeding cues (*Figure 8F*). Although BKOs exhibited prolonged pre-feeding activity in DD (*Figure 7F*; *Figure 8C*), only the liver and kidney showed tightly clustered phases (*Figure 8G*; *Figure 8—source data 1A*). Other tissues (heart, lung, spleen) were variable in phase, though their phase distribution was different either from that found in an *ad libitum* condition (see also *Figure 5D*) or from those in Fx/Fx (*Figure 8G*, *Figure 8—source data 1A*). These results suggest that feeding cues differentially entrain peripheral clocks; FR strongly synchronizes circadian clocks in the liver and kidney but has weaker effects on other peripheral tissues such as the pituitary, heart, lung, and spleen.

To our surprise, with the exception of the liver, oscillatory amplitude of peripheral rhythms of BKOs remained lower than that of Fx/Fx control mice even after 2 weeks of FR (*Figure 8I*). Lower amplitude is not related to the state of phase synchrony, because relative amplitude of kidney still remained low even though FR synchronized its phase. This indicates that the damped rhythms were not completely restored by food signals.

To compare *ad libitum* and FR conditions directly, we performed a 'meta-analysis' on phase and amplitude within Fx/Fx and BKOs, respectively (*Figure 9A–D*). Statistical comparison between *ad libitum* in DD (BKO from DD) and FR in DD (BKO from DD + FR) confirmed tissue specificity of phase entrainment by feeding (*Figure 9*; *Figure 9—source data 1A,B*). This result is in striking contrast to the case in which BKOs were exposed to LD cycles, which synchronized all peripheral rhythms examined even in the absence of a functioning SCN (*Figure 9B,D*; *Figure 9—source data 1B*). Nevertheless, both internal synchrony and oscillatory amplitude in peripheral tissues were maintained most optimally when an intact SCN was present (*Figure 5*; *Figure 6A*; *Figure 8H*). Taken together, these results indicate that, while individual tissues can be entrained by light/dark cycles and feeding schedules independently, the central clock plays an important role in sustaining coherence among peripheral tissues and robust circadian programs at the organismal level.

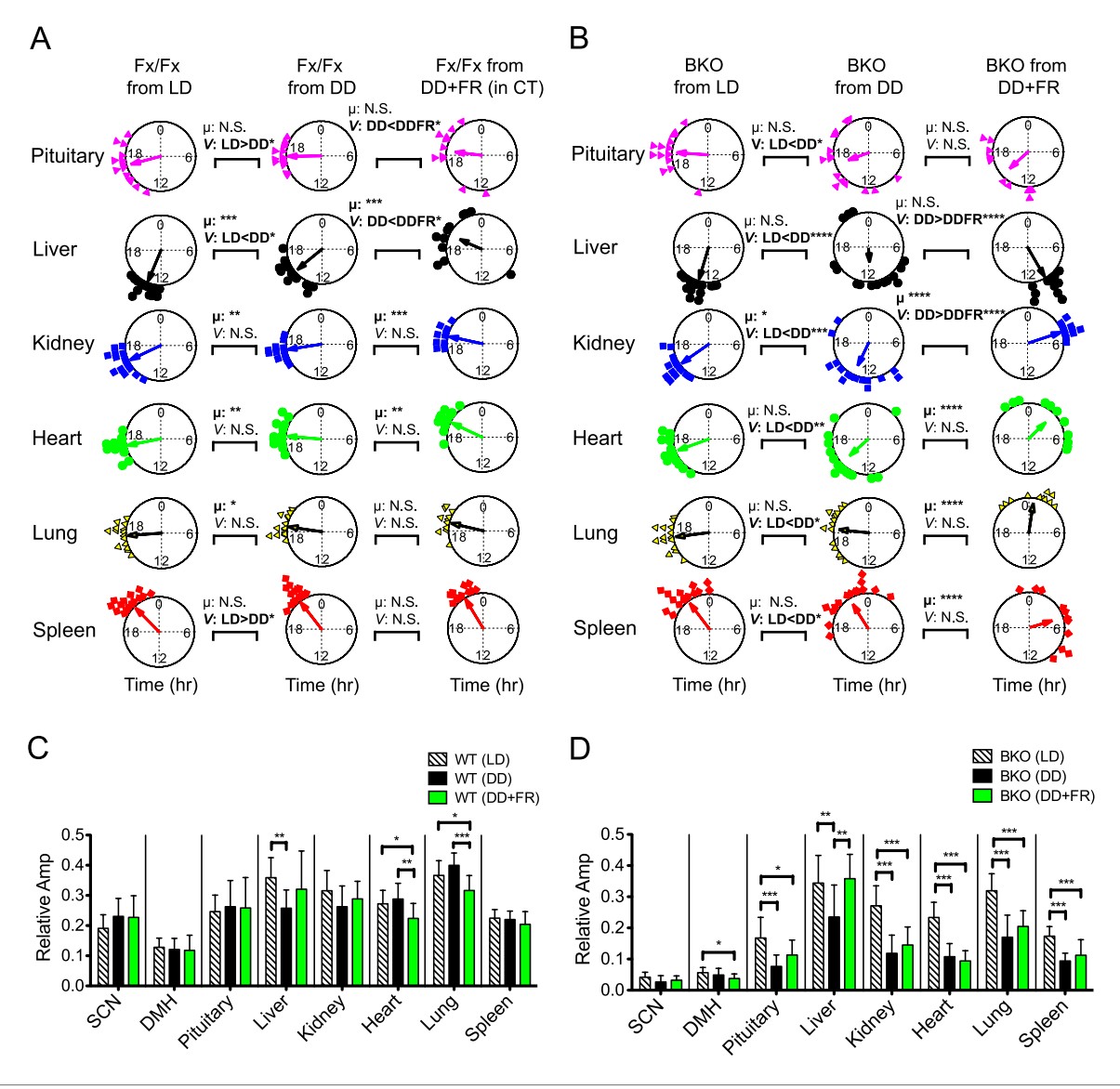

**Figure 9**. Analysis of circadian rhythms of peripheral tissues under different conditions. (**A**, **B**) Comparison of peak bioluminescence rhythms in the peripheral tissues in Fx/Fx (**A**) and BKO (**B**) mice under LD, DD, and FR conditions. Circular plots are as presented in *Figure 5E* and *Figure 8G*. Watson-Williams F-test and bootstrap analysis were performed to compare the mean phase angle (μ) and the variance (*V*, distribution of peak phase values) between two circular data, respectively. Statistical comparison was summarized in *Figure 9—source data 1A,B*). *p < 0.05, **p < 0.01, ***p < 0.001, ****p < 0.0001. N.S. = not significant. (**C**, **D**) Comparison of normalized amplitude of bioluminescence rhythms from Fx/Fx (**C**) and BKO (**D**) mice under LD, DD, and FR conditions. Each bar graph is as presented in *Figure 6A,B* and *Figure 8I*. Mean relative amplitude ±SD are shown. *p < 0.05, **p < 0.01, ***p < 0.001 by ANOVA.

The following source data is available for figure 9:

**Source data 1**. (**A**) Summary of statistical comparison of peak phases from Fx/Fx mice under different conditions. (**B**) Summary of statistical comparison of peak phases from BKO mice under different conditions.

## Discussion

### Circadian behavioral rhythmicity is abolished by deleting *Bmal1* from the forebrain

*Bmal1* is the only component of the mammalian circadian clock whose sole deletion generates a complete loss of circadian rhythms in mice (*Bunger et al., 2000*). Therefore, elimination of *Bmal1* from

a target tissue provides an invaluable model to study the behavioral and physiological consequences of abolishing circadian rhythms in a tissue-specific manner. In this study, we succeeded in removing *Bmal1* specifically from the forebrain using a *Camk2a::iCre*BAC line (*Casanova et al., 2001*). Notably, this *CamiCre* driver was able to efficiently recombine and delete our target floxed gene in the entire SCN. These forebrain/SCN-specific *Bmal1* knockouts exhibited a total loss of circadian behavioral rhythmicity, and their circadian phenotype in DD and LL is similar to that of global *Bmal1* knockouts (*Bunger et al., 2000*). Thus, *Bmal1* in the forebrain is essential for normal circadian activity rhythms. However, a Cre driver that can express more specifically in the SCN would be necessary to confirm the dominance of the SCN over other sites of the brain.

BKOs displayed apparent entrainment to LD cycles; however, additional experiments revealed two deficits in these mice. First, entrainment in a skeleton photoperiod was abnormal with a bimodal pattern. A similar pattern is found in pinealectomized sparrows in which a population of oscillators entrains with two different and opposite phase angles (*Takahashi and Menaker, 1982*). Second, there is a delay in masking behavior to light under LD 3.5:3.5 which is similar to that seen in *Clock* mutants (*Redlin et al., 2005*). Thus, the variable phase angle of entrainment in BKOs on LD 12:12 is consistent with weaker synchronization of residual oscillators as well as a small deficit in masking.

## Impact of a loss of a functional forebrain clock on circadian organization of peripheral tissues

Despite a complete loss of circadian behavioral rhythmicity, *Bmal1* forebrain-specific knockout mice are as healthy as wild-type mice, consistent with our earlier results using tissue-specific rescue of *Bmal1⁻/⁻* mice (*McDearmon et al., 2006*). These results suggest that *Bmal1*'s contribution to the circadian phenotype seen in *Bmal1⁻/⁻* mice is distinct from its role in the maintenance of normal activity levels, body weight, reproduction, and longevity. For instance, the skeletal muscle defects seen in *Bmal1⁻/⁻* mice (*Andrews et al., 2010*) are likely a causative factor in their low activity. Muscle-rescued *Bmal1⁻/⁻* mice exhibit improvement of body weight, activity, and longevity but not of arthropathy (*McDearmon et al., 2006*). Our findings as well as others' suggest that the effects of BMAL1 can be dissociated between brain and other tissues including the reproductive system.

While the effect of *Bmal1* deletion is local, the impact of disabling the brain's clock on peripheral circadian organization is significant. Previous studies have investigated the effect of SCN ablation on peripheral rhythms by making SCN lesions and recording circadian gene expression of peripheral tissues either ex vivo (*Yoo et al., 2004*) or in vivo (*Tahara et al., 2012*; *Saini et al., 2013*). Here, we used a Cre-mediated genetic excision approach to remove a critical component of the molecular oscillator from the intact SCN network. This strategy demonstrates that internal synchrony and high amplitude rhythms are lost in the absence of a functional master oscillator in the forebrain/SCN and supports previous reports (*Yoo et al., 2004*; *Tahara et al., 2012*; *Saini et al., 2013*). Nevertheless, further studies are needed to elucidate the functional consequences of damped circadian rhythms in each peripheral organ.

On the other hand, the dispersed phases in BKOs were not observed when mice were placed in an LD condition (*Figure 5C,E*). Given that the BKO's circadian behavior is driven by light from retinal pathways presumably acting via the SCN, this demonstrates that cyclic photic input (*Figure 3*) is sufficient to sustain coherent temporal synchrony of peripheral oscillators. Alternatively, the effects of light could also be mediated by activity (wheel-running)-associated stimuli driven by LD as another source of systemic signals impinging on peripheral clocks analogous to that found previously (*Kornmann et al., 2007*; *Hughes et al., 2012*).

## Forebrain *Bmal1* expression is not necessary for food entrainment behavior

BKOs offer an advantage for investigating the effects of a disabled master clock in the brain because they do not suffer from the morbid phenotypes observed with a global loss of *Bmal1*. Using the forebrain *Bmal1*-deficient animals, we were able to conduct food restriction experiments under completely standard conditions. In contrast to global *Bmal1* knockouts, altered experimental conditions such as enriched liquid food (*Storch and Weitz, 2009*), LD 18:6 (*Pendergast et al., 2009*), or food placed on the cage bottom, and gentle handling to prompt feeding (*Fuller et al., 2008*) were not required for these mice. Our results show that BKOs indeed displayed normal FAA as other studies using global *Bmal1* knockouts have reported (*Mistlberger et al., 2008*; *Pendergast et al., 2009*; *Storch and Weitz, 2009*), demonstrating that BMAL1 in the forebrain is not essential for food entrainment behavior.

Specifically, our *Bmal1* in situ results in the DMH indicate that the DMH is not necessary for FAA as reported previously (*Landry et al., 2006*, *2007*; *Moriya et al., 2009*; *Acosta-Galvan et al., 2011*; *Landry et al., 2011*). In our experiments, BKOs tend to exhibit intact food-anticipatory behavior under constant darkness conditions. However, overall activity of BKO's circadian behavior was also significantly elevated in DD, which makes it difficult to distinguish whether the elevated pre-feeding activity is due to enhanced FAA or to the effect of overall activity increase. Nevertheless, these results suggest that an inhibitory system (*Davis and Menaker, 1980*) suppresses the total activity of both circadian behavior and FAA. It remains to be determined what factors underlie suppression of the activity and how their regulation in the SCN and other brain regions is altered by the deletion of BMAL1.

## Differential effects of light/dark cycles and feeding cues

Previous work has shown that restricted feeding can alter the phase of peripheral rhythms without affecting the phase of the SCN (*Damiola et al., 2000*; *Stokkan et al., 2001*). While peripheral oscillators can be uncoupled from the central clock in the SCN, recent studies have suggested that SCN-derived signals counteract the feeding cues for rapid phase-shifting of peripheral clocks (*Saini et al., 2013*) and that phase-shifting rates are variable among different tissues (*Damiola et al., 2000*; *Yamazaki et al., 2000*). Consistent with these observations, peripheral tissues in our control mice displayed variable phases relative to the FR schedule when conducted in constant darkness (*Figure 8D*). In these free-running conditions, the circadian activity rhythms of the mice were not affected by FR; and as a consequence, the phase relationships between the FR schedule and circadian phase could be dissociated. When control mice in DD under *ad libitum* conditions were compared to control mice in DD under FR using circadian phase as a marker, the liver, kidney and heart significantly shifted their mean circular phase in response to feeding, showing an effect of FR on these tissues [*Figure 9A*, compare Fx/Fx from DD with Fx/Fx from DD + FR (in CT); *Figure 9—source data 1A*]. However, the phases of the peripheral rhythms were significantly clustered with the phase of the activity rhythm (*Figure 8E*), strongly suggesting that signals from the SCN continue to dictate the expression of peripheral phases in wild-type mice despite the expression of FAA behavior. Indeed, only the liver was strongly clustered in phase with FR in DD (*Figure 8G*, Fx/Fx from DD + FR) compared to the circadian phase of activity [*Figure 8G*, Fx/Fx from DD + FR (in CT)] in which all peripheral tissues were clustered in phase. Thus we see a rather complex picture in control mice in which FR exerts tissue-specific effects on peripheral organ circadian phase and coherence.

In contrast, in the absence of a functional pacemaker in the forebrain, liver and kidney clocks shifted to a distinct phase in response to food restriction and became synchronized (*Figure 8F,G*; *Figure 9B*), which supports the results of an in vivo study using SCN-lesioned animals (*Saini et al., 2013*). We found, however, that other tissues such as heart, lung, and spleen did not entrain to food restriction even after 2 weeks of treatment (*Figure 9B*; *Figure 9—source data 1A*). Although the precise mechanism is not known, the lower degree of synchrony of these tissues could be due to the longer duration of pre-feeding behavioral activity which could mobilize additional (conflicting) signals. At the same time, these tissues showed distinct phase shifts in response to feeding (*Figure 9B*; *Figure 9—source data 1B*). Circular phase variance analysis indicates that internal coherence among different organs within an individual mouse still remained disrupted (*Figure 8H*). Therefore, these results suggest that, while synchrony of individual tissues such as liver and kidney can be maintained independently by a feeding schedule, synchrony among peripheral tissues requires central clocks in the forebrain/SCN.

The effects of FR were different from the effects of LD cycles (*Figure 5*; *Figure 9B*), which sustained both phase synchrony and oscillatory amplitude in peripheral clocks in BKOs (*Figure 9*). Notably, both phase angle (*Figure 5E*; Figure 5—source data 1A) and phase coherence among peripheral tissues (*Figure 5F*) were similar to those in Fx/Fx control mice which possess an intact central clock. We speculate that, while feeding cues can antagonize SCN-derived signals, LD cycles appear to act in concert with the SCN or signals arising from the SCN. Although detailed pathways through which the SCN controls the peripheral clocks remain unknown, recently, SRF was identified as a transcription factor that regulates gene expression in peripheral tissues in response to systemically oscillating signals (*Gerber et al., 2013*). It will be interesting to explore how these signals affect circadian gene expression in vivo to determine the mechanisms by which the SCN communicates with peripheral oscillators. Our experiments reveal the diverse behavioral and peripheral physiological consequences of inactivating a forebrain clock. They pave the way for further exploring the complex relationship between central and peripheral clocks.

## Materials and methods

### Animals

All mice were housed under LD 12:12 unless otherwise noted. *Camk2a::iCre*BAC mice (*CamiCre*) (MGI:2181426) were kindly provided by Dr G. Schutz (*Casanova et al., 2001*). For a Cre reporter mouse, Rosa26-CAG-LSL-tdTomato (Ai14, kindly provided by Dr Hongkui Zeng) was used (*Madisen et al., 2010*). *Bmal1*fx/fx mice (*Johnson et al., 2014*) are homozygous for a *Bmal1* allele which has *loxP* sites inserted into the introns surrounding exon 4. *Bmal1*fx/fx mice (129SvJ × C57BL/6J N3 backcross) were crossed to *CamiCre* to produce *Bmal1*fx/fx (Fx/Fx), *CamiCre+;Bmal1*fx/+ (Het) , and *CamiCre+; Bmal1*fx/fx (BKO) mice. *CamiCre* mice were mated to their siblings (FVB/N × C57BL/6J N3 backcross) to generate *CamiCre* hemizygous controls. *Bmal1*−/− global knockout mice (*Bunger et al., 2000*) were 129SvJ × C57BL/6J N14 backcross congenic animals. For the real-time reporting assay, PER2::LUC mice (*Yoo et al., 2004*) were crossed to *Bmal1*fx/fx to produce *CamiCre+; Bmal1*fx/fx; *Per2*Luc/+ and its control littermate, *Bmal1*fx/fx; *Per2*Luc/+. For the bioluminescence imaging experiments, *Per2*Luc mice were homozygous. For all experiments, male and female mice were used in balanced ratios; mice were 2–7 months old for behavior experiments and 3–11 months old for bioluminescence recordings and age-matched across groups, except for FR (males only were used for behavior experiments), where mice were 1.9–3.6 months old at the start of the experiment. All animal studies and Materials and methods were in accordance with Northwestern University and UT Southwestern Medical Center guidelines for animal care and use.

### PCR genotyping

The following sets of genotyping primers were used: *CamiCre*: iCre-PCR-F 5′-TCTGATGAAGTCA GGAAGAACC-3′ and iCre-PCR-R 5′-GAGATGTCCTTCACTCTGATTC-3′ (amplified a 400 bp product). PCR reactions were carried out in a final volume of 30 μl buffer consisting of 5 μl Promega 5× Colored buffer, 2.7 μl 25 mM MgCl$_2$, 0.38 μl 20 μM each primer, 3 μl 2 mM dNTPs, 3 μl NaOH-extracted tail genomic DNA, and 0.3 μl Promega Go Taq (1 U/rx; Promega, Madison, WI). PCR conditions were: 5 min at 95°C, followed by 30 cycles of 1 min at 94°C, 2 min at 58°C, and 2 min at 72°C, followed by 8 min at 72°C. The reaction products were analyzed on a 2% agarose gel. *Bmal1*fx/fx: OL5436 5′- CCC TGA ACA TGG GAA AGA GA -3′ and OL6013 5′- ATT CAC CTT TTG GGG AGG AC -3′ (floxed band: 360 bp, WT band: 310 bp). PCR reactions were carried out in a final volume of 25 μl buffer consisting of 5 μl Promega 5× Colored buffer, 3 μl 25 mM MgCl$_2$, 1 μl 20 μM each primer, 2.5 μl 2 mM dNTPs, 3 μl NaOH-extracted tail genomic DNA, and 0.3 μl Promega Go Taq (1.25 U/rx). PCR conditions were: 5 min at 95°C; followed by 37 cycles of 30 s at 95°C, 30 s at 60°C, and 30 s at 72°C, followed by 5 min at 72°C. Ai14: Ai14F 5′- TAC GGC ATG GAC GAG CTG TAC AAG TAA -3′and Ai14R 5′- CAG GCG AGC AGC CAA GGA AA -3′ (amplified a 517 bp product). *Bmal1*−/− and PER2::LUC mice were genotyped as previously described (*Bunger et al., 2000*; *Yoo et al., 2004*).

### Behavioral experiments and activity monitoring

To record the rhythm of locomotor activity, adult mice (at least 8-week old) were individually housed in activity wheel-equipped cages under LD 12:12 for at least 10 days. Locomotor activity was recorded and analyzed using ClockLab software (Actimetrics, Wilmette, IL) as previously described (*Vitaterna et al., 1999*; *Siepka and Takahashi, 2005*; *McDearmon et al., 2006*; *Vitaterna et al., 2006*). Fluorescent lights (300–600 lux inside the cage) were used for behavior experiments. Mice were transferred into DD for 4 weeks, returned to LD for 2 weeks, released into LL for 4 weeks, and then returned to LD for at least 2 weeks. For the entrainment experiment, mice were placed in LD 12:12 for at least 10 days, released into DD for 4 weeks, followed by LD for 2 weeks, before they were introduced to a skeleton photoperiod (LDLD 1:10:1:12) for 4 weeks. The mice were returned to LD for 2 weeks and then switched to an ultradian cycle (LD 3.5:3.5) for 3 weeks.

### Food restriction experiments

For the food restriction experiment, mice were placed in wheel-equipped cages under LD 12:12 for 2 weeks, then underwent a 5-day FR ramp of gradually decreasing food availability (*Figure 7A*), followed by 14 days of FR in LD. They were returned to *ad lib* feeding under LD for 2 weeks, DD for 2 weeks, and LD for 2 weeks. Then the mice underwent a 5-day FR ramp after which they were released into DD for 14 days of FR in DD. On the first day of the FR ramp, food was removed at ZT/CT18 and 12 hr later food was returned. On each successive day, food was removed 2 hr earlier until food was

available for 4 hr, from ZT/CT6–10 (*Figure 7A*). This was the final food availability window for 14 days of FR. A regular mouse diet (5K52; Lab Diet, Richmond, IN; contains 19.3% protein and 6.2% fat, or 2918; Irradiated Global Diet, Teklad Diets, Madison, WI; contains 18.6% protein and 6.2% fat) was used during FR experiments. Food pellets were weighed daily for each mouse before and after food availability to monitor food consumption. Individual mean daily activity profiles were computed by ClockLab; group average profiles were made using Microsoft Excel.

For the bioluminescence recording experiment, mice were initially entrained in LD 12:12 for a minimum of 6 days (6–12 days), as the light-on time was delayed for 6 hr, before they were released into DD for 2 weeks. Then the mice underwent a 5-day FR ramp as above, after which 4-hr FR was conducted from ZT/CT6–10 for 2 weeks (*Figure 8A*). On the final day of 2 weeks of FR, tissues were harvested for bioluminescence recordings at 2 hr after the food was withdrawn (ZT/CT12).

## Behavioral data analysis

Wheel-running activity was recorded and analyzed essentially as previously described (*Vitaterna et al., 2006*). The free-running periods in DD and LL were calculated from the entire 28-day interval by using $\chi^2$ periodogram analysis (Clocklab software, Actimetrics). The amplitude of circadian rhythm was analyzed using the fast Fourier transform (FFT), which estimates the relative power of approximately 24 hr period rhythm in comparison with all other periodicities in the time-series (*Shimomura et al., 2001*). The power spectral densities for frequencies ranging from 0 to 1 cycle/hr were determined and normalized to a total power (area under the curve) of 1.0. The peak in the circadian range (18- to 30-hr period or 0.033–0.055 cycles/hr) of the relative power was determined for each animal for comparison. If no significant periodicity in the 18–30 hr range was detected by FFT, the free-running period was not scored. For analysis of total daily activity, the total number of wheel revolutions per day was averaged in LD (last 10 days of initial LD interval), DD, and LL (28 days) using the same days described above for free-running period and amplitude analysis. Effects of genotype were analyzed by a Generalized Model (GLM) ANOVA by using NCSS (Kayesville, UT), with Tukey–Kramer multiple comparison post-tests for pairwise comparisons.

Two-sample t-tests were used to analyze subjective day vs night activity during skeleton photoperiod and amplitudes during LD 3.5:3.5, and Repeated Measures GLM ANOVA with Tukey–Kramer multiple comparison post-tests was used to compare the time course of masking between genotypes during days 3–15 of the 21-day LD 3.5:3.5 cycle. FFT was used to analyze amplitudes in the circadian as well as 7-hr range during LD 3.5:3.5. To analyze FR data, Repeated Measures GLM ANOVA with Tukey–Kramer multiple comparison post-tests was used to compare the effect of genotype, time, and interaction between the two on FAA (absolute counts), FAA (% of total activity), weights, total daily activity, and food intake. Two-sample t-test and paired t-test, respectively, were used to calculate FAA onset and activity fold-change from *ad lib* to FR.

## In situ hybridization

For the time course collection of brain tissue for in situ hybridization, after 2 weeks of entrainment in LD, animals were released into DD for 2 days and tissues were harvested on the third day of DD. Brains were rapidly dissected and frozen on dry ice and stored at −80°C until further processing. Standard in situ hybridization procedures were used as previously described (*Sangoram et al., 1998*). Alternate 20-μm thick coronal sections were collected from each brain from the rostral to the caudal end of the SCN or DMH (24–32 total, 12–16 for each probe) by using a Leica CM3050S cryostat. Sections were thaw-mounted onto SuperFrost plus microscope slides (VWR, Radnor, PA), then stored at −80°C until all sectioning was completed. Alternate sections were hybridized to a combination oligoprobe against *mBmal1* exon 4 and to a riboprobe against *Per2*. For the *Bmal1* combination oligoprobe, three 37 to 50-mer oligonucleotide probes (IDT, Coralville, IA) were radiolabeled at the 3′ ends with [33]P via terminal I deoxynucleotidyl transferase (Gibco/Invitrogen, Life Technologies, Grand Island, NY) and used in equal proportions: Mop3Ex4Probe1 5′-AACTGTTCATTTTGTCCCGACGCC TCTTTTCAATCTGACTGTGGGCCTCC-3′, Mop3Ex4Probe2 5′-CCTGGACATTGCATTGCATGTTGGTAC CAAAGAAGCCAATTCATC-3′, Mop3Ex4Probe3 5′-CTGAACAGCCATCCTTAGCACGGTGAGTTTATC TAAC-3′. Templates for the anti-sense and sense *Per2* probes were PCR-generated by using the following primers: Per2-insitu-f 5′-ACG AGA ACT GCT CCA CGG GAC-3′, Per2-insitu-r 5′-ACA GCC ACA GCA AAC ATA TCC GC-3′. PCR products were cloned into a TA cloning vector (Invitrogen). The T7 promoter from the TA cloning vector was used to generate both anti-sense and sense probes.

All probes were labeled with $^{33}$P. Quantitation of the autoradiogram signal was performed by using NIH ImageJ 1.34s software (NIH, Bethesda, MD) and normalized to radioactive standards as described previously (*Vitaterna et al., 1999*). The optical density (OD) of individual SCN or DMC was normalized by subtracting the OD of an area of identical size in the lateral hypothalamus or the magnocellular nucleus of lateral hypothalamus, respectively, from the same side (left or right) and section. Normalized values from three sections near the middle (anterior–posterior) SCN or caudal DMH (at the level of DMC) were used to calculate an average for each brain. Effects of genotype, time, or their interactions were analyzed by GLM ANOVA, with Tukey–Kramer multiple comparison post-tests for pairwise comparisons.

## Western blotting

For Westerns blots, tissues were harvested at ZT16 from mice in LD 12:12. Tissues were homogenized by a Polytron homogenizer in a buffer containing 44.6 mM Tris–HCl, 5.5 mM Tris–Base, 154 mM NaCl, 29.8 mM sodium pyrophosphate, 20 mM glycerol 2-phosphate, 50 mM NaF, 1 µg/ml aprotinin, 1 µg/ml leupeptin, 1 mM PMSF, 1 mM sodium orthovanadate, 1 mM EDTA, 1 mM p-nitrophosphate, 0.1% SDS, 0.5% sodium deoxycholate, and 1% NP-40. Homogenates were centrifuged at 15,000×$g$ for 20 min at 4°C. Supernatants were collected and protein concentration was estimated using Bio-Rad DC Protein Assay according to the manufacturer's instructions. Total protein (10 µg) was diluted 1:1 with Laemmli sample buffer and resolved on a 10% SDS-polyacrylamide gel by electrophoresis. Thereafter, proteins were electrotransferred onto a GE Healthcare PVDF transfer membrane. The membranes were blocked with PBST (PBS + 0.1% Tween-20) containing 5% non-fat powdered milk for 1 hr and then incubated with the rabbit polyclonal anti-MOP3 antibody (1:6,250, generated in Dr Bradfield's lab and is available at NB100-2288, Novus Biologicals LLC, Littleton, CO), followed by anti-rabbit IgG secondary antisera horseradish peroxidase (1:1000; PI-1000, Vector Laboratories, Burlingame, CA). Proteins were visualized with a chemiluminescence detection system (ECL Western blotting detection analysis system; Amersham Pharmacia, GE Healthcare Bio-Sciences, Pittsburgh, PA) and with subsequent exposure to autoradiographic films.

## Immunohistochemistry and microscopy

Brain tissues for immunohistochemistry were harvested at ZT16 under LD 12:12. Animals were anesthetized with ketamine/xylazine/saline cocktail (10 mg/ml ketamine, 10 mg/ml xylazine) at 0.01 ml/g body weight or with sodium pentobarbital at 120 mg/kg body weight and then perfused intracardially with 25 ml of 4% paraformaldehyde (pH 7.4, Sigma-Aldrich, St Louis, MO) in 63.4 mM phosphate buffer (pH 7.5). The brains were removed and post-fixed for 2 hr at 4°C in 4% paraformaldehyde in 63.4 mM phosphate buffer and transferred to 20% sucrose/phosphate buffer overnight. For immunohistochemistry, 50-µm coronal sections were collected through the entire SCN using a Leica cryostat and processed free-floating. Sections were incubated with the rabbit polyclonal anti-MOP3 (BMAL1) antibody (1:1000) followed by anti-rabbit IgG secondary biotinylated antisera (1:200; BA-1000, Vector Laboratories) and visualized using Vectastain ABC Kit and 0.5 mg/ml diaminobenzidine, 0.01% hydrogen peroxide, and 0.03% NiCl$_2$ in 0.05 M Tris (pH 7.2). Sections were mounted onto gelatin-coated microscope slides using aqueous mounting medium (3:1 glycerol/phosphate buffer), cover-slipped, and imaged with a Leica DM-RB upright microscope using Openlab software in the Biological Imaging Facility at Northwestern University. For the whole brain imaging, a Zeiss Stereo Discovery Microscope V12 was used under a bright field. Fluorescence images were taken using an Olympus MVX10 (1×) and a Zeiss LSM 510 confocal microscope (20× objective).

## Real-time bioluminescence recording and data analysis

To monitor real-time reporting of PER2::LUC oscillations, tissues were harvested after cervical dislocation between ZT/CT10.5–13. Coronal sections of the brain were sliced at 300-µm thickness by a vibratome in ice-cold Hank's balanced salt solution (HBSS, Invitrogen). Each individual SCN was dissected out and cultured on a Millicell organotypic membrane (PICM ORG50, Millipore, Billerica, MA) in a 35 mm tissue culture dish containing 1.2 ml DMEM (90-013-PB, Mediatech, Manassas, VA), supplemented with 2% B27 (Invitrogen), 10 mM Hepes (pH 7.2), 4 mM L-glutamine, 0.035% sodium bicarbonate, 25 units/ml penicillin, 25 µg/ml streptomycin, and 0.1 mM luciferin (L-8240, Biosynth AG, Staad, Switzerland). Other tissues were processed as previously described (*Yamazaki and Takahashi, 2005*) and cultured as above. The bioluminescence was monitored by a light-tight 32-channel LumiCycle

luminometry (Actimetrics, Wilmette, IL) maintained at 36°C and was recorded at an interval of 10-min continuously for a minimum of 8 days followed by medium change to confirm the viability of the samples.

All data were analyzed essentially as described previously (*Izumo et al., 2006*). Briefly, raw data of bioluminescence records were corrected for background counts and PMT gain. For animals harvested from a DD condition (30–44 days in DD), activity onset of each Fx/Fx control mouse was set as CT12. The time-series data were detrended by a 24-hr running average and then subjected to FFT-NLLS analysis to calculate periods and phases. The initial 20 hr of data were trimmed because they contain an acute effect of explant preparation. Data beyond 192 hr before medium change were also trimmed except for one group (1 Fx/Fx control and 1 BKO), which was affected by an electrical shutdown on day 6. The amplitude was computed at a 36–60 hr range and normalized as previously described (*Izumo et al., 2006*). For all data, the calculations were averaged where duplicated samples were prepared from the same tissue within an individual mouse, and the averaged value was used to map or calculate for all mice collected. Samples with RelAmp Error ≥0.84 (background counts) or out of a circadian (17–31 hr) range (24 hr ± 30% of 24 hr) in the periodicity were excluded from the analysis. A total of 1106 LumiCycle files were processed and used in this study. To construct a phase map, smoothed time-series were used to determine the peak on the second day. Phase angles and circular variances of circular plots were computed using Oriana (Kovach Computing Services, Wales, UK). The variance of the peak phases between two groups was compared using bootstrap analysis (*Sato et al., 2007*). For each iteration of the bootstrap, the difference of the variance between the two groups was calculated by random sampling with replacement from the data within the each of the two groups. Following 20,000 iterations, ninety-five percent confidence intervals were used to determine whether the differences were significantly different from zero.

## Bioluminescence imaging and data analysis

For real-time bioluminescence imaging analysis, age-matched pairs of mice (Fx/Fx control and BKO) were placed in DD for >30 days before harvesting the tissues. The tissue collection was repeated for a total of three pairs of mice. The tissues were processed and prepared in the same manner as in the luminometry recording, except that a glass bottom culture dish (MatTek, Ashland, MA) was used. The sealed culture was placed onto the stage chamber maintained at 36°C inside the LV200 Bioluminescence Imaging System (Olympus America, Irving, TX). The sample was imaged using a 10× objective lens with 10-min exposure time at 25-min interval for 7 days. The imaging files were converted to linear time-series by quantifying the signals on grid matrices. The heat map was generated using the top 200 time-series data starting with the strongest signals. The top 50 of these time-series data were analyzed for period, phase, and relative amplitude by FFT-NLLS as described above. The phases calculated by FFT-NLLS were further converted to the actual time and divided by each circadian period to normalize period differences. Statistical analysis was conducted between paired mice as described above.

## Acknowledgements

We thank Ms Amanda Falk for help with mouse production, Dr Tereza Smejkalova for help with Westerns, Dr William Russin at the Biologial Imaging Facility and Dr Xinran Liu for help with imaging, Dr David Ferster for help with data analysis, Mr Giuseppe Fruci for assistance with figure production, Drs Martha Vitaterna, Hee-Kyung Hong, Marleen de Groot, Kazuhiro Shimomura, and Ethan Buhr for helpful discussions. We also thank Dr Hongkui Zeng at Allen Institute for Brain Science for Ai14 mice. JST is an investigator in the Howard Hughes Medical Institute.

## Additional information

### Funding

| Funder | Grant reference number | Author |
| --- | --- | --- |
| Howard Hughes Medical Institute | H0690101 | Joseph S Takahashi |
| National Institute of Mental Health | U01 MH61915, P50 MH074924 | Joseph S Takahashi |

| Funder | Grant reference number | Author |
|--------|------------------------|--------|
| National Institute of Neurological Disorders and Stroke | 1F30NS056551-01 | Martina Pejchal |
| Brain and Behavior Research Foundation | | Mariko Izumo |
| National Institute of Environmental Health Sciences | R01 ES005703 | Christopher A Bradfield |
| Japan Science and Technology Agency | | Takashi R Sato |

The funders had no role in study design, data collection and interpretation, or the decision to submit the work for publication.

### Author contributions

MI, MP, Conception and design, Acquisition of data, Analysis and interpretation of data, Drafting or revising the article; ACS, RPL, Acquisition of data, Analysis and interpretation of data; JAW, TRS, Analysis and interpretation of data, Contributed unpublished essential data or reagents; XW, CAB, Drafting or revising the article, Contributed unpublished essential data or reagents; JST, Conception and design, Analysis and interpretation of data, Drafting or revising the article

### Author ORCIDs

Joseph S Takahashi, http://orcid.org/0000-0003-0384-8878

### Ethics

Animal experimentation: All animal care and use procedures were in accordance with guidelines of the Northwestern University (Protocol 2006-0035) and UT Southwestern Institutional Animal Care and Use Committees (Protocols 2009-0054 and 2012-0090).

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
