## [Decision Letter]

Thank you for sending your work entitled “Light But Not Feeding Rescues Temporal Organization of Peripheral Circadian Clocks in a Brain Bmal1 Conditional Mutant” for consideration at *eLife*. Your article has been favorably evaluated by a Senior editor, a Reviewing editor, and 3 reviewers.

The following individuals responsible for the peer review of your submission have agreed to reveal their identity: Louis Ptacek (Reviewing editor); Ralph Mistlberger (peer reviewer). Two other reviewers remain anonymous.

The Reviewing editor and the other reviewers discussed their comments before we reached this decision, and the Reviewing editor has assembled the following comments to help you prepare a revised submission.

The primary hypothesis being addressed in this paper is that temporal organization of peripheral clocks may be maintained *in vivo* by light and/or feeding cues, when the master clock in the suprachiasmatic nuclei is genetically altered. For that purpose, the authors generated forebrain Bmal1 conditional mutant mice (noted KO thereafter). In constant darkness, these mice are behaviorally arrhythmic, while their peripheral clocks were still oscillating, albeit in a desynchronized way as compared to control mice.

1) The authors consider that light, but not feeding, rescues temporal organization of peripheral clocks in KO mice. The data provided suggest alternative interpretations. Because the temporal organization of peripheral clocks in KO mice exposed to a light-dark cycle (LD) matches remarkably that in SCN-functional control mice (Figure 5), this indicates that the direct effects of light cues given by LD are sufficient to maintain temporal organization in peripheral clocks (i.e., light cues do not rescue any defect in KO mice). Based on the immediate behavioral arrhythmicity after transfer to DD, while the rhythmicity is apparent under regular LD, there is no obvious reason to speak of weak entrainment to light in KO mice. The apparent rhythmicity under LD is likely due to direct effects of daily exposure to 12 hours of light per cycle. A recent study using Syt10(Cre) driver to alter the clock in the SCN has shown that timing in a peripheral clock (i.e., adrenal gland) can be maintained by exposure to LD (Husse et al. 2014 FASEB J). If really the authors consider that light rescues temporal organization of peripheral clocks in KO mice without involving masking effects of light, they should prove it by investigating such temporal organization in KO mice exposed to lighting conditions that maintain entrainment of the suprachiasmatic nuclei, while limiting the masking effects of light, such as a skeletal photoperiod (as used here for behavioral measurements shown in Figure 3) or daily 1 h light pulses followed by 23 h of darkness.

2) Feeding cues in constant darkness are said not to rescue temporal organization of peripheral clocks in KO mice. However, visual inspection of Figure 8 suggests that phase dispersion between peripheral clocks is not different, if not improved, in KO mice (Figure 8, panel F) as compared to control fx/fx mice (Figure 8, panel D). The authors' conclusion is based on data converted in circadian times for control mice, and compared to data expressed according to Zeitgeber (feeding) time in KO mice. Even with data conversion, phase dispersion in KO mice is not different for the pituitary, liver and kidney. The question here is to investigate the synchronizing effects of feeding in DD and because it is not possible to convert data of KO mice in circadian times, data from both fx/fx and KO mice should be compared according to feeding time. The results will probably lead to the conclusion that feeding cues have synchronizing effects in KO mice, at least not worst, and maybe better than in control mice with functional SCN. Together, these points suggest that both light and feeding maintain temporal organization of peripheral circadian clocks in forebrain Bmal1 conditional mutant mice.

Some conclusions were felt to be too strong and restatement of these is necessary, as follows.

3) A major result and substantive conclusion of this study, as stated in the abstract, is that “Light cycles, but not restricted feeding treatment, rescued this defect [loss of synchrony of peripheral clocks, between and within tissues, in brain bmal1 deficient mice, BKOs], revealing the dominance of photic entrainment pathways acting via the SCN over nutrient signals”. This conclusion is summarized in the Discussion section, by statements that light appears to be a stronger entraining signal than feeding cues for most peripheral clocks, and that the SCN plays a 'critical' role in actively sustaining internal synchronization. I would argue that this representation of the results is not quite accurate, and that the conclusion is overstated, for the following reasons:

a) The midday restricted feeding schedule did fully rescue the clock in the liver (both phase and amplitude), and rescued phase in the kidney. Figure 8 also shows that phase in the heart, lung, pituitary and spleen are at least partly rescued, compared to the adlib DD condition. The statement of the results in the Abstract should be revised to acknowledge that loss of synchrony was rescued by restricted feeding in the liver and kidney, but only partially in the other tissues. Synchrony among peripheral tissues requires central bmal1 dependent clocks, while synchrony of individual tissues to a feeding schedule may not require central clocks.

b) The statement that light appears to be stronger than feeding for 'most' peripheral clocks might give the impression that most peripheral tissues were examined. If the stomach, duodenum, colon and pancreas had been included, perhaps 'most' of the clocks would have been found to be rescued, i.e., the conclusion reversed. Selection of the tissues is no doubt important, and worth emphasizing.

c) The statement that light appears to be 'stronger' than feeding in phase control of peripheral clocks is not supported by the data, because the study wasn't designed to compare relative strengths. Relative strength has already been established by studies of intact animals; when food is restricted to the daytime, peripheral clocks uncouple from LD and SCN and shift to align with mealtime. There may be conditions under which light is more effective than feeding, for some tissues (possibly the minority, pending further study), but references to relative strength of food and light should be avoided, or stated in a more nuanced fashion.

4) The statement that the SCN plays a 'critical' role in actively sustaining internal synchronization is also debatable, because Bmal1 was absent from most of the rest of the brain, and some of the loss of amplitude or phase control in peripheral tissues, particularly in the restricted feeding condition, could be due to loss of inputs from Bmal1-dependent FEOs in the brain outside of the SCN. Presumably, hypothetical bmal1-dependent FEOs (in PVN, DMH, arcuate, brainstem?) communicating with peripheral organs would have to be distinct from FEOs responsible for food anticipatory behavioral rhythms, given that the behavioral rhythms were quite robust in the BKOs.

5) The conclusion that cyclic photic input is sufficient to sustain coherent temporal synchrony of peripheral oscillators is supported by the data, but the mechanism by which light does this remains an open question. The LD cycle in BKO mice drives a daily rhythm of activity and presumably feeding, and the effects of light may be mediated by stimuli associated with these behaviors. Indeed, it would not be surprising if the heart and lung circadian clocks are regulated by various systemic effects of wheel running (exercise); the phase of those clocks may be regulated by the timing of behavior. FAA in DD was much longer in duration in the BKO mice, and that might explain the lower degree of synchrony of heart and lung clocks during restricted feeding in those mice. I would recommend briefly noting this as a potential mechanism.

---

## [Author Response]

*1) The authors consider that light, but not feeding, rescues temporal organization of peripheral clocks in KO mice. The data provided suggest alternative interpretations. Because the temporal organization of peripheral clocks in KO mice exposed to a light-dark cycle (LD) matches remarkably that in SCN-functional control mice (*Figure 5*), this indicates that the direct effects of light cues given by LD are sufficient to maintain temporal organization in peripheral clocks (i.e., light cues do not rescue any defect in KO mice). Based on the immediate behavioral arrhythmicity after transfer to DD, while the rhythmicity is apparent under regular LD, there is no obvious reason to speak of weak entrainment to light in KO mice. The apparent rhythmicity under LD is likely due to direct effects of daily exposure to 12 hours of light per cycle. A recent study using Syt10(Cre) driver to alter the clock in the SCN has shown that timing in a peripheral clock (i.e., adrenal gland) can be maintained by exposure to LD (Husse et al. 2014 FASEB J). If really the authors consider that light rescues temporal organization of peripheral clocks in KO mice without involving masking effects of light, they should prove it by investigating such temporal organization in KO mice exposed to lighting conditions that maintain entrainment of the suprachiasmatic nuclei, while limiting the masking effects of light, such as a skeletal photoperiod (as used here for behavioral measurements shown in*
Figure 3*) or daily 1 h light pulses followed by 23 h of darkness*.

We agree with the reviewers’ comment that LD cycles are likely the primary stimulus maintaining temporal organization of peripheral clocks in BKO mice. It is not our intention to say that masking effects of light are not involved. To avoid confusion and also to incorporate the comments below, we have modified the title to, “Differential effects of light and feeding on circadian organization of peripheral clocks in a forebrain *Bmal1* mutant” and emphasized “light/dark cycles” in the revised manuscript.

*2) Feeding cues in constant darkness are said not to rescue temporal organization of peripheral clocks in KO mice. However, visual inspection of*
Figure 8
*suggests that phase dispersion between peripheral clocks is not different, if not improved, in KO mice (*Figure 8*, panel F) as compared to control fx/fx mice (*Figure 8*, panel D). The authors' conclusion is based on data converted in circadian times for control mice, and compared to data expressed according to Zeitgeber (feeding) time in KO mice. Even with data conversion, phase dispersion in KO mice is not different for the pituitary, liver and kidney. The question here is to investigate the synchronizing effects of feeding in DD and because it is not possible to convert data of KO mice in circadian times, data from both fx/fx and KO mice should be compared according to feeding time. The results will probably lead to the conclusion that feeding cues have synchronizing effects in KO mice, at least not worst, and maybe better than in control mice with functional SCN. Together, these points suggest that both light and feeding maintain temporal organization of peripheral circadian clocks in forebrain Bmal1 conditional mutant mice*.

We agree with the reviewers and thank them for their careful consideration of our description. In an attempt to oversimplify our description of the results, we overgeneralized our conclusions. We have now described the results in more detail to emphasize the differential effects of FR on peripheral rhythms. We have performed additional analysis of the phase data of Fx/Fx and BKO tissues by comparing the peak phases plotted against feeding time ([Supplementary-material SD2-data], Fx/Fx ZT vs BKO). The statistical analysis indeed showed a significance shift in the mean phase angle in kidney, heart, lung, and spleen (but not in pituitary and liver) in BKOs. Comparison of the circular variance also showed that the phases of BKO’s were more clustered in liver, kidney, lung, and spleen (but not in pituitary and heart) as compared to Fx/Fx, suggesting that feeding cues have synchronizing effects on peripheral clocks in the absence of functional SCN.

This analysis as well as additional analyses that we present in a new Figure 9 ([Supplementary-material SD2-data], Figure 9, and [Supplementary-material SD3-data] as described below), together suggest that both light (in the form of LD cycles) and feeding have an effect on the temporal organization of peripheral oscillators in the forebrain *Bmal1* conditional knockout mice. However, we also noted that their effects are not equal. While the synchronizing effect of light is similar to the effect from the SCN (as manifested in the DD condition), the restricted feeding shifts the phases of peripheral rhythms differently and clusters the phase variance only in liver and kidney in BKO’s. To aid in the visualization of all results in one page, we performed a “meta-analysis” of all phase and amplitude data taken under different conditions and summarized the analysis in Figure 9 and [Supplementary-material SD3-data]. This analysis illustrates the differential effects of light and feeding more clearly.

*Some conclusions were felt to be too strong and restatement of these is necessary, as follows*.

*3) A major result and substantive conclusion of this study, as stated in the abstract, is that “Light cycles, but not restricted feeding treatment, rescued this defect [loss of synchrony of peripheral clocks, between and within tissues, in brain bmal1 deficient mice, BKOs], revealing the dominance of photic entrainment pathways acting via the SCN over nutrient signals”. This conclusion is summarized in the Discussion section, by statements that light appears to be a stronger entraining signal than feeding cues for most peripheral clocks, and that the SCN plays a 'critical' role in actively sustaining internal synchronization. I would argue that this representation of the results is not quite accurate, and that the conclusion is overstated, for the following reasons*:

*a) The midday restricted feeding schedule did fully rescue the clock in the liver (both phase and amplitude), and rescued phase in the kidney.*
Figure 8
*also shows that phase in the heart, lung, pituitary and spleen are at least partly rescued, compared to the adlib DD condition. The statement of the results in the Abstract should be revised to acknowledge that loss of synchrony was rescued by restricted feeding in the liver and kidney, but only partially in the other tissues. Synchrony among peripheral tissues requires central bmal1 dependent clocks, while synchrony of individual tissues to a feeding schedule may not require central clocks*.

We have revised the title, Abstract, and other text in the manuscript by specifying which tissues were synchronized by restricted feeding. Please note, however, that the statistical analysis for Figure 9 (which includes Figure 8) did not reveal a difference in the phase dispersion in the heart, lung, spleen, and pituitary between the adlib DD and FR condition ([Supplementary-material SD3-data], which was previously numbered as [Supplementary-material SD2-data]): i.e., the phases of these tissues remained dispersed.

Also, we performed analysis on the circular variance among different tissues within individual mice that underwent restricted feeding (current [Supplementary-material SD2-data]). BKO’s still show a significant phase variation between organs (new Figure 8), supporting the hypothesis that “synchrony among peripheral tissues requires central *Bmal1* dependent clocks, while synchrony of individual tissues to a feeding schedule may not require central clocks”. Based on this additional analysis, we have modified our Results and Discussion to state this point.

*b) The statement that light appears to be stronger than feeding for 'most' peripheral clocks might give the impression that most peripheral tissues were examined. If the stomach, duodenum, colon and pancreas had been included, perhaps 'most' of the clocks would have been found to be rescued, i.e., the conclusion reversed. Selection of the tissues is no doubt important, and worth emphasizing*.

We agree and have revised relevant sentences by emphasizing the selection of the tissues used in the study instead of making a sweeping statement.

*c) The statement that light appears to be 'stronger' than feeding in phase control of peripheral clocks is not supported by the data, because the study wasn't designed to compare relative strengths. Relative strength has already been established by studies of intact animals; when food is restricted to the daytime, peripheral clocks uncouple from LD and SCN and shift to align with mealtime. There may be conditions under which light is more effective than feeding, for some tissues (possibly the minority, pending further study), but references to relative strength of food and light should be avoided, or stated in a more nuanced fashion*.

Based also on the comments and our responses in item #2, we altered the phrases from “relative strengths” to “differential effects” of light and feeding where applicable in the revised manuscript.

*4) The statement that the SCN plays a 'critical' role in actively sustaining internal synchronization is also debatable, because Bmal1 was absent from most of the rest of the brain, and some of the loss of amplitude or phase control in peripheral tissues, particularly in the restricted feeding condition, could be due to loss of inputs from Bmal1-dependent FEOs in the brain outside of the SCN. Presumably, hypothetical bmal1-dependent FEOs (in PVN, DMH, arcuate, brainstem?) communicating with peripheral organs would have to be distinct from FEOs responsible for food anticipatory behavioral rhythms, given that the behavioral rhythms were quite robust in the BKOs*.

We have deleted this statement in the Discussion. We agree that it is not possible to speculate about the roles of other brain regions at this time, given that our conditional *Bmal1* knockout is not specific to the SCN. Therefore, we state that “a Cre driver that can express more specifically in the SCN would be necessary to confirm the dominance of the SCN over other sites of the brain.”

*5) The conclusion that cyclic photic input is sufficient to sustain coherent temporal synchrony of peripheral oscillators is supported by the data, but the mechanism by which light does this remains an open question. The LD cycle in BKO mice drives a daily rhythm of activity and presumably feeding, and the effects of light may be mediated by stimuli associated with these behaviors. Indeed, it would not be surprising if the heart and lung circadian clocks are regulated by various systemic effects of wheel running (exercise); the phase of those clocks may be regulated by the timing of behavior. FAA in DD was much longer in duration in the BKO mice, and that might explain the lower degree of synchrony of heart and lung clocks during restricted feeding in those mice. I would recommend briefly noting this as a potential mechanism*.

We have incorporated this point as a potential mechanism in Discussion in the revised manuscript.